# Social training reconfigures prediction errors to shape Self-Other boundaries

Sam Ereira [1,2✉], Tobias U. Hauser [1,2], Rani Moran [1,2], Giles W. Story[1,2], Raymond J. Dolan [1,2] & Zeb Kurth-Nelson[1,3]

Selectively attributing beliefs to specific agents is core to reasoning about other people and imagining oneself in different states. Evidence suggests humans might achieve this by simulating each other's computations in agent-specific neural circuits, but it is not known how circuits become agent-specific. Here we investigate whether agent-specificity adapts to social context. We train subjects on social learning tasks, manipulating the frequency with which self and other see the same information. Training alters the agent-specificity of prediction error (PE) circuits for at least 24 h, modulating the extent to which another agent's PE is experienced as one's own and influencing perspective-taking in an independent task. Ventromedial prefrontal myelin density, indexed by magnetisation transfer, correlates with the strength of this adaptation. We describe a frontotemporal learning network, which exploits relationships between different agents' computations. Our findings suggest that Self-Other boundaries are learnable variables, shaped by the statistical structure of social experience.

[1] Max Planck UCL Centre for Computational Psychiatry and Ageing Research, UCL, London WC1B 5EH, UK. [2] Wellcome Centre for Human Neuroimaging, UCL, London WC1N 3BG, UK. [3] DeepMind, London N1C 4AG, UK. ✉email: samuel.ereira.14@ucl.ac.uk

Humans tend to align their beliefs and values with each other[1–4], particularly if they are part of a common social group[5–7]. Adopting another agent's beliefs facilitates social integration and enables exploitation of the other agent's knowledge about an environment[8]. However, representing another agent's beliefs without adopting those beliefs is equally crucial for predicting the behaviour of others and engaging in fluid social interactions[9–12]. It is necessary then to strike a context-dependent balance between Self-Other distinction on the one hand and Self-Other mergence on the other.

Humans revise their beliefs about the world by computing prediction error (PE) signals, which compare internal predictions with actual experiences of the environment[13]. There is evidence suggesting that the brain can also simulate another agent's predictive signals. Simulated reward prediction errors (RPEs) have been observed in the medial prefrontal cortex (mPFC) of humans engaged in observational learning[14,15], explicit mentalising[16], social teaching[17] and prosocial learning[18]. Beyond the domain of reward learning, simulations of another agent's sensory surprise signals, have been observed in humans, using both functional magnetic resonance imaging (fMRI)[19] and magnetoencephalography (MEG)[20].

When simultaneously tracking one's own beliefs and another agent's beliefs, the brain computes sensory PEs for each agent using distinct agent-specific neural circuits, where the degree of neural distinction predicts a behavioural Self-Other distinction[20]. This means that neural representations of PEs contain information, not only about events taking place in the environment, but also about the identity of the agent who is modelling that environment. If Self-Other distinction is indeed achieved by representing computations in agent-specific circuits, one should expect the agent-specificity of these circuits to adapt to changes in social context. In other words, a high-level learning process should enable the brain to reconfigure low-level learning signals to be either more or less agent-specific, as a function of prior social experience.

Here, we examine whether a Self-Other distinction is susceptible to experience-dependent plasticity by training subjects on a mentalising task with two social contexts. We show, using computational modelling and fMRI, that when Self and Other share a high number of concurrent experiences, there is a sustained increase in the neural overlap between Self-attributed and Other-attributed PEs. Conversely, when Self and Other share a low number of concurrent experiences, there is a sustained reduction in the neural overlap between Self-attributed and Other-attributed PEs. This training manipulation also modifies Self-Other distinction in a separate transfer task that does not involve learning. We use quantitative MRI to show that the myeloarchitecture of ventromedial prefrontal white matter is associated with this relearning of Self-Other boundaries and we also find evidence that the ventromedial prefrontal cortex (vmPFC) directly tracks the probability of sharing experiences with another agent. Finally, we present results that suggest the mechanisms through which Self-Other boundaries are acquired are also used in the non-social context of intertemporal reasoning.

## Results

**Experimental set up**. We trained subjects on a probabilistic false belief task (FBT)[20] with two social contexts (Fig. 1a). On each trial, subjects received a sample from a Bernoulli distribution with a parameter, $p$, that drifted across trials. They were tasked with periodically predicting $p$ or another player's estimate of $p$, by reporting a corresponding value on a continuous probability scale. The other player had a false belief about $p$ because they received a corrupted stream of information (Fig. 1b), as follows.

On 'privileged' trials, the Bernoulli sample was visible only to the subject. On 'shared' trials, the sample was visible to both players. On 'decoy' trials, a false sample was delivered to the other player, which the subject could see and knew was misleading. This design results in belief trajectories for Self and Other that are essentially uncorrelated (Fig. 1c and Methods). The other player was a real person, playing a simplified version of the game (Methods).

Subjects played two separate games, each game with a different other player, depicted by a distinct cartoon avatar. During training with a 'Hi-Share' avatar, 50% of trials were 'shared'. With a 'Lo-Share' avatar 12.5% of trials were 'shared'. During testing, 24 h later and concurrent with functional magnetic resonance imaging (fMRI), subjects played with both agents again, experiencing now 1/3 'shared' trials with each agent. We predicted that subjects would simulate the PEs of the other agent, but also that the degree of experience-sharing in training would determine the extent to which Self- and Other-attributed PEs were neurally segregated. Thus, we predicted that, at test, Self-Other distinction would be greater in the Lo-Share context than in the Hi-Share context.

**Behavioural adaptation of Self-Other distinction**. We quantified performance by correlating subjects' predictions of the Bernoulli parameter, $p$, with the true $p$ used to generate the observed outcomes (Fig. 2a). We were interested in whether any behavioural differences would persist into the testing session, where the Hi-Share and Lo-Share tasks were statistically identical. Indeed, in the testing session subjects performed significantly worse in the Hi-Share context [repeated measures ANOVA: $F(1, 39) = 6.76$, $p = 0.013$]. We also observed an interaction between context (Hi- or Lo-Share) and probe trial (Self or Other) on performance [repeated measures ANOVA: $F(1, 39)$, $p = 0.023$]. Specifically, performance in the Hi-Share context was impaired on Self probe trials [paired t-test: $t(39) = 2.96$, $p = 0.005$] but not on Other probe trials [paired t-test: $t(39) = 0.15$, $p = 0.89$].

We fit learning models (see Methods) to subjects' choice behaviour in order to assess whether disrupted Self-Other distinction provided an explanation for the impaired performance in the Hi-Share context. For example, a belief ($B$) update for Self could be implemented as follows:

$$B_{t+1} = B_t + \alpha\text{PE}^{\text{self}} + \delta(0.5 - B_t) + \lambda\text{PE}^{\text{other}} \qquad (1)$$

Here a participant's belief, $B_{t+1}$, about the Bernoulli parameter is given by their belief on the previous trial, $B_t$, plus a prediction error, $\text{PE}^{\text{self}}$, based on Self-relevant information seen during the trial, weighted by a learning rate, $\alpha$. The third term implements a memory decay, whereby beliefs drift towards chance level with rate $\delta$. The final term captures an inability to completely segregate learning for Self and Other, such that belief updates for Self are sensitive to the Other's prediction error, $\text{PE}^{\text{other}}$, governed by a leak parameter, $\lambda$. A parallel belief update is also implemented for the simulation of the Other's learning. We tested variant models of the form above with separate parameters for Self and Other, or for 'shared' and 'privileged'/'decoy' trials (see Supplementary Table 2). All models had at least one learning rate ($\alpha$) and at least one decision temperature ($\tau$). Some models included 1 or 2 memory decay parameters ($\delta$) and some models included 1 or 2 Self-Other leak parameters ($\lambda$).

Behaviour in the Hi-Share context, in both training and testing, was best explained by learning models that included a $\lambda$ parameter, whilst behaviour in the Lo-Share context was best explained by models without $\lambda$. This parameter adds an additional update to one agent's belief that reflects the irrelevant agent's PE[20]. Thus, a Self-attributed PE is also used to update an Other-attributed belief and vice versa, merging the belief

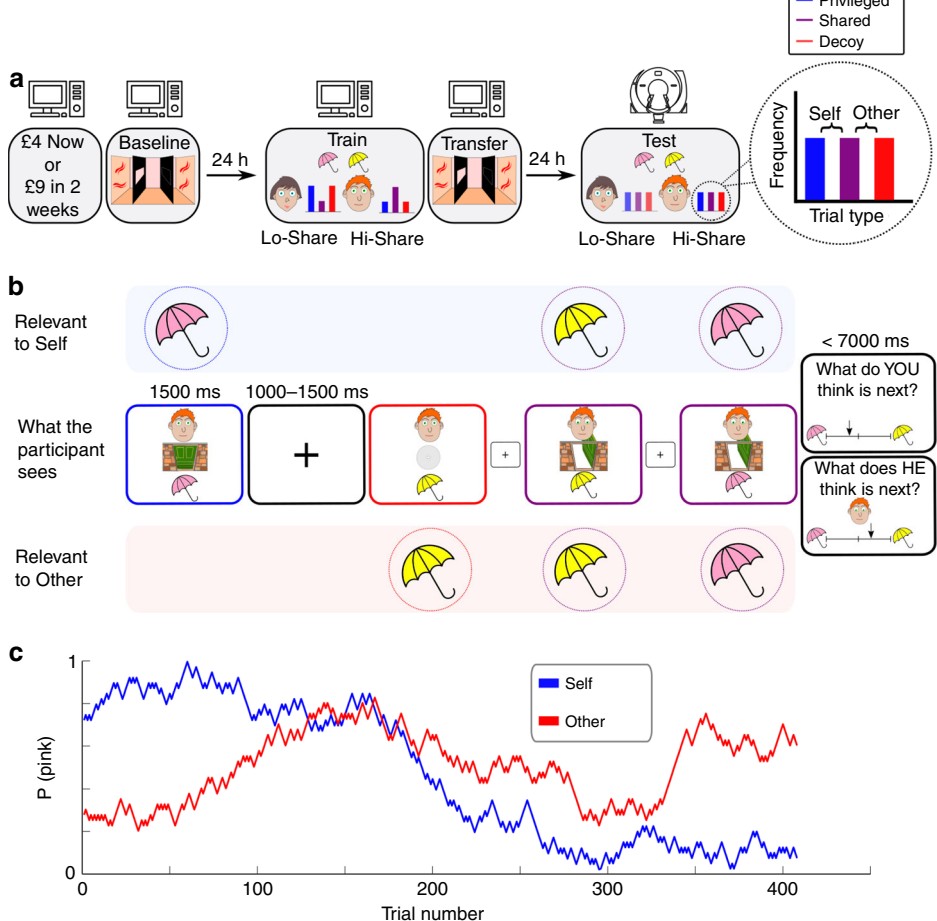

**Fig. 1 Experimental design. a** Three-day experimental timeline. On day 1 subjects played an intertemporal choice task followed by a visual perspective-taking task (see Fig. 3a). On day 2 subjects were trained on the false belief task shown in **b**, with two different social contexts. Here, the Lo-Share context is represented by the female avatar and the Hi-Share context is represented by the male avatar. Subjects then played the visual perspective-taking task again to measure transfer effects. On day 3 subjects were tested on the false belief task with concurrent fMRI. This time there were no statistical differences between the two social contexts. **b** Trial structure of the probabilistic false belief task. The middle row shows what the subject sees on the computer display. Each trial comprises a Bernoulli outcome on the bottom half of the display (pink or yellow), and an image on the top half of the display, which indicates whether the trial is 'privileged' (blue), 'shared' (purple), or 'decoy' (red). Subjects were intermittently probed to report their estimate of the Bernoulli parameter, $p$ (Self-probe), or their estimate of the other agent's false belief about $p$ (Other-probe). **c** An example pair of random walks used to generate a trial sequence for the false belief task. The trial sequence is designed to induce uncorrelated beliefs in Self and Other.

trajectories for Self and Other (Methods, Supplementary Figs. 1–3). The model that best explained behaviour in the Hi-Share task in testing contained a single $\lambda$ parameter, but multiple learning rates ($\alpha$) for different trial types. We found that $\alpha$ was higher for learning on behalf of Other than for Self [paired t-test: t(39) = −2.27, $p$ = 0.029] and therefore $\lambda$ made a stronger contribution to Self-updates than Other-updates. The difference in $\lambda$:$\alpha$ ratio for Self and Other in the Hi-Share task, in testing, negatively correlated with the difference in performance between Self and Other probe trials [Spearman's rank correlation: $\rho$ = −0.45, $p$ = 0.004] and so may partially explain the interaction shown in Fig. 2a.

Fig. 2b shows a Self-Other merging effect induced by $\lambda$, by visualising the correlation between model-derived Self- and Other-attributed belief trajectories. A main effect of context on Self-Other correlation was observed in both training [repeated measures ANOVA: F(1, 39) = 21.7, $p$ < 0.001] and testing [repeated measures ANOVA: F(1, 39) = 3.8, $p$ < 0.001], with higher Self-Other correlation in Hi-Share than Lo-Share. By simulating belief trajectories using different values of $\lambda$ we observed that $\lambda$ was a strong determinant of this Self-Other merging effect (Supplementary Fig. 3). The learning

models approximated the empirical data well and the parameters were identifiable (Fig. 2a, c, d). The different types of model were identifiable in a recovery analysis (Supplementary Fig. 4).

**Behavioural training transfers to a perspective-taking task.** We next examined whether this training effect on Self-Other distinction generalises to a different cognitive domain. Subjects were exposed to an adapted version of a visual perspective-taking task[21] before and after FBT training (Fig. 3a). Subjects were required to process visual scenes either on behalf of Self or of avatars who had restricted visual perspectives. All trials started by asking the subject to adopt a perspective, either Self or Other. This was followed by presenting the subject with a target pattern and number. Finally, subjects were presented with a visual scene with an avatar in the centre who saw only half of the scene. The scene contained four patterns, some of which were identical to the target pattern and some of which were distractors. Subjects had three seconds to give a binary response, indicating whether or not they saw the target number of target patterns, if adopting the relevant perspective, Self or Other.

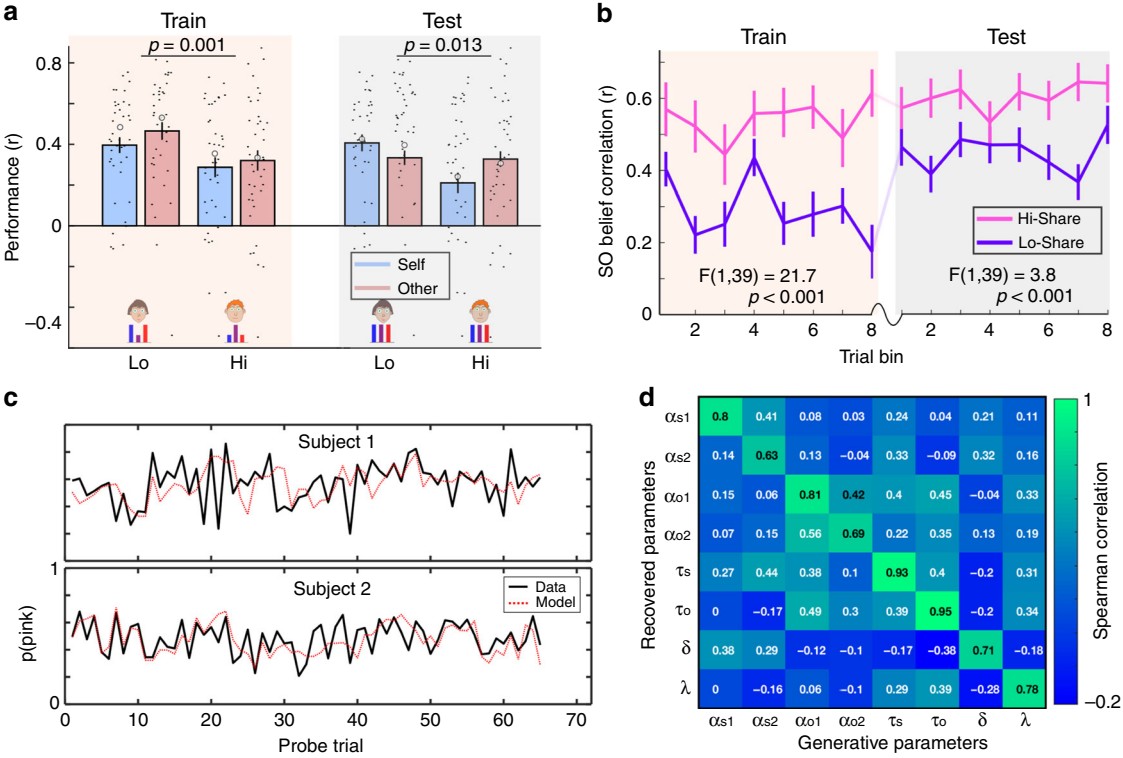

**Fig. 2 Behavioural training induces sustained changes in Self-Other distinction ability. a** Performance in FBT, split by session (train or test), context (Lo-Share or Hi-Share) and probe trial (Self or Other). In both sessions, there was a main effect of context with worse performance in the Hi-Share context. Large white circles show the behaviour predicted by the winning models. $n = 40$ independent subjects. Repeated measures ANOVA testing for main effect of condition in training session $F(1, 39) = 12.64$, $p = 0.001$. Repeated measures ANOVA testing for main effect of condition in testing session $F(1, 39) = 6.78$, $p = 0.013$. **b** Correlation between model-derived Self- and Other-attributed beliefs in different trial bins, split by session and context. In both sessions, there was a main effect of context with a higher Self-Other correlation in the Hi-Share context (pink line) than in the Lo-Share context (purple line). This finding was invariant to the number of trial bins used in the analysis (see Supplementary Fig. 5) and was driven largely by the $\lambda$ parameter (Supplementary Figs. 2 and 3). Repeated measures ANOVA testing for main effect of condition in training session $F(1, 39) = 21.7$, $p < 0.001$. Repeated measures ANOVA testing for main effect of condition in testing session $F(1, 39) = 3.8$, $p < 0.001$. **c** Generative performance of the best-fitting model for the Hi-Share context in the test session for two exemplar subjects. Self and Other probe trials are intermixed in the order they were presented to the subjects. **d** Parameter identifiability of the most complex winning model (Hi-Share context test session). Each cell shows a Spearman correlation coefficient derived from correlating subjects' true parameter estimates with recovered parameter estimates. A subscript 's' indicates that the parameter is specific to Self updates. A subscript 'o' indicates that the parameter is specific to Other updates. A subscript '1' indicates that the learning rate is specific to 'privileged' and 'decoy' trials. A subscript '2' indicates that the learning rate is specific to 'shared' trials. All error bars denote s.e.m. Source data are provided as a Source Data file.

Typically, in such tasks, subjects perform poorly if the avatar's visual perspective of the scene is incongruent with the subject's[21]. Our adapted task alternated between trials with the avatar from the Hi-Share context of the FBT and trials with the avatar from the Lo-Share context. Thus, subjects were required to distinguish between the visual perspectives of Self and multiple Others. Furthermore, in a third of trials, instead of seeing an avatar, subjects saw an arrow in the centre of the scene. On these trials, subjects were required to indicate whether or not the arrow was pointing to the target number of target patterns. These 'arrow' trials, enabled us to quantify FBT training-related changes, over and above non-specific repeat effects that were unrelated to FBT training (Methods).

We quantified behaviour with drift rate parameters from a drift–diffusion model fit to response time and accuracy data (see Methods), where a higher drift rate indicates faster and more accurate responses[22]. Empirical and simulated data are shown in Supplementary Fig. 6. We found higher drift rates on congruent compared to incongruent trials (Fig. 3b) at both baseline [paired t-test: t(45) = 7.6, $p < 0.001$] and transfer [paired t-test: t(45) = 6.7, $p < 0.001$]. We also observed a main effect of avatar on corrected drift rate change (Methods) from baseline to transfer

[repeated measures ANOVA: F(1, 45) = 7.4, $p = 0.009$], with performance improving on trials with the Lo-Share avatar and worsening on trials with the Hi-Share avatar (Fig. 3c), consistent with increased and reduced Self-Other distinctions, respectively.

The training task (FBT) required subjects to attribute learning signals to Self and Other but the transfer task required subjects to attribute visual perspectives to Self and Other, with no learning involved. These results suggest that during FBT training, subjects learned about the relationship between computations of Self and Other, in a manner specific to the identity of the other agent, but invariant to the computations being attributed to that agent.

**Adaptation of neural Self-Other distinction.** In addition to behavioural effects, our hypothesis predicts that the neural segregation of Self- and Other-attributed computational signals should change with experience. We first localised unsigned PE signals in the brain, collapsing over the Hi-Share and Lo-Share contexts. Using both mass-univariate and searchlight multi-voxel pattern analyses (Methods), we found Self- and Other-attributed sensory PEs in extrastriate, parietal and supplementary motor cortices (Fig. 4a, Supplementary Fig. 7, Supplementary Table 1).

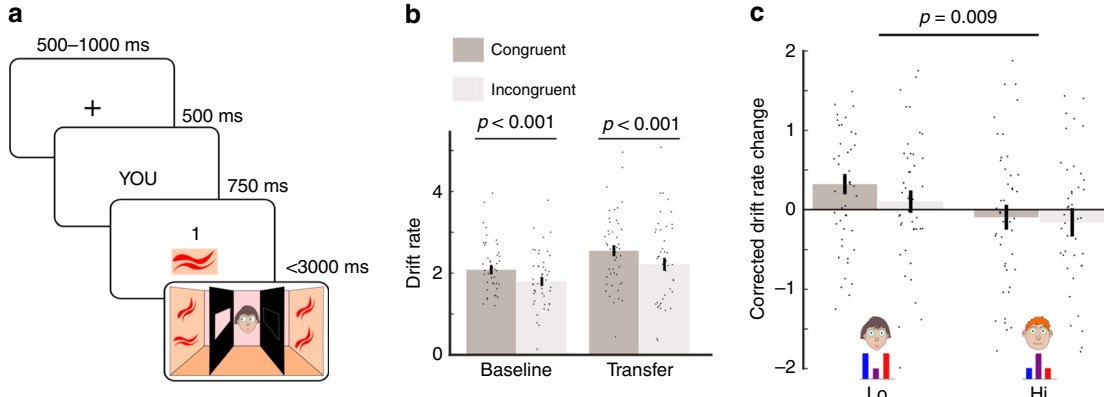

**Fig. 3 Behavioural training transfers to a perspective-taking task. a** Trial structure of the perspective-taking task. Subjects were first presented with a perspective (Self or Other), followed by a target pattern and number and then a visual scene. Subjects had three seconds to report whether or not the scene contained the target number of target patterns, if adopting the relevant perspective. **b** Incongruency effect in the perspective-taking task, averaged over Lo-Share and Hi-Share. Mean drift rate parameters were higher on congruent compared to incongruent trials at both baseline and transfer phases. $n =$ 46 independent subjects. Baseline: Paired two-sided t-test: t(45) = 7.6, $p < 0.001$. Transfer: Paired two-sided t-test: t(45) = 6.7, $p < 0.001$. **c** Transfer effect in the perspective-taking task. 'Corrected drift rate change' (see Methods) was significantly higher on trials with the Lo-Share agent than trials with the Hi-Share agent. $n =$ 46 independent subjects. Repeated measures ANOVA testing for main effect of avatar: F(1, 45) = 7.4, $p = 0.009$. Error bars denote s.e.m. Source data are provided as a Source Data file.

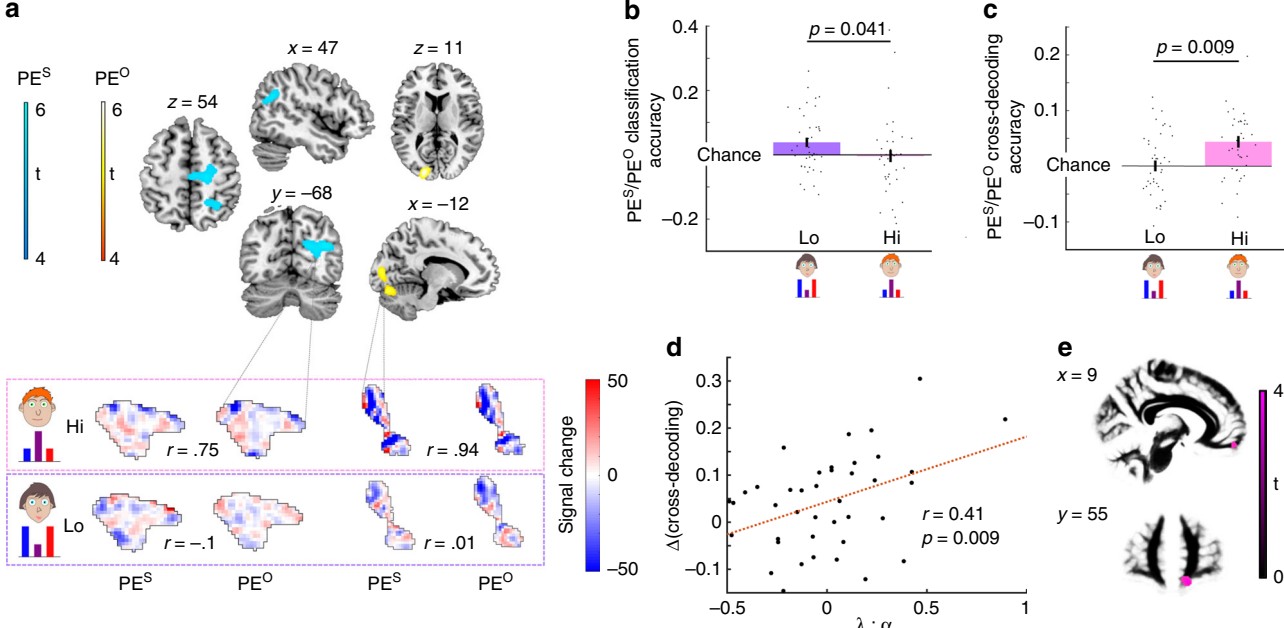

**Fig. 4 Representations of PEs adapt with behavioural training. a** Clusters of voxels where BOLD signal covaried with |PE|, either Self-attributed (PE^s) or Other-attributed (PE^o), from a searchlight analysis, using two one-sided t-contrasts on $n = 40$ independent subjects. Clusters were defined with a cluster-forming threshold of $p < 0.001$. Only clusters large enough to survive FWE-correction at $p < 0.05$ are displayed (top). BOLD signal was extracted from these clusters and patterns of PE-related activity were compared for Self- and Other-attributed signals for the Lo-Share and Hi-Share contexts. Exemplar trial patterns are shown from a single subject. The 'signal change' refers to the difference in BOLD signal between a trial with a large PE and a trial with a small PE (see Methods). Patterns for PE^self and PE^other are more similar for the Hi-Share context than the Lo-Share context (bottom). **b** Decoding performance (cross-entropy below chance) when classifying PE activity patterns as Self- or Other-attributed. Classification accuracy was significantly higher in the Lo-Share context than the Hi-Share context. $n = 40$ independent subjects. Paired two-sided t-test: t(39) = 2.1, $p = 0.041$. **c** Decoding performance when predicting |PE^other| after training on |PE^self| and vice versa. Cross-decoding accuracy (Fisher Z-transformed correlation) was significantly higher in the Hi-Share context than the Lo-Share context. $n = 40$ independent subjects. Paired two-sided t-test: t(39) = 2.75, $p = 0.009$. **d** The difference in cross-decoding performance shown in **c**, is positively correlated with the ratio of leak to learning rate (parameters derived from the Hi-Share context). This indicates a relationship between neural Self-Other mergence and behavioural Self-Other mergence. $n = 40$ independent subjects. Pearson correlation: $r = 0.41$, $p = 0.009$. **e** Cluster of voxels where myelin-related MT covaried with the difference in cross-decoding shown in **c**, overlaid on white matter. Clusters were defined with a cluster-forming threshold of $p < 0.001$. Only clusters that were large enough to survive FWE-correction at $p < 0.05$ are displayed. All error bars denote s.e.m. Source data are provided as a Source Data file.

We defined a mask as the union of these clusters. Within this multi-cluster mask we trained classifiers on activity patterns of Self- and Other-attributed PEs and tested their ability to classify PEs from left-out trials, as being Self- or Other-attributed (Methods). Self-Other classification was significantly better than chance in the Lo-Share context [one-sample t-test: $t(39) = 2.6$, $p = 0.013$] but not in the Hi-Share context [one-sample t-test: $t(39) = -0.26$, $p = 0.8$], with a significant difference in classification accuracy between the two contexts [paired t-test: $t(39) = 2.1$, $p = 0.041$] (Fig. 4b). This finding is consistent with the activity patterns for Self- and Other-attributed PEs being distinct in the Lo-Share context and overlapping in the Hi-Share context.

To exclude the possibility that the null result in Hi-Share did not reflect overlapping activity patterns, but instead reflected noisier data, we tested for the logical inverse of the classification analysis. Here, we directly tested for similarity of Self- and Other-attributed PE representations by training linear regression models on patterns of Self-attributed PEs and testing them on patterns of Other-attributed PEs, and vice versa. As shown in Fig. 4c, we found that cross-decoding accuracy was significantly better than chance in the Hi-Share context [one-sample t-test: $t(39) = 4.22$, $p < 0.001$] but not in the Lo-Share context [one-sample t-test: $t(39) = 0.06$, $p = 0.95$], with a significant difference in cross-decoding performance between the two contexts [paired t-test: $t(39) = 2.75$, $p = 0.009$]. These findings are consistent with FBT-training inducing plasticity in the neural representations of PEs, such that Self-Other mergence of PE signals was promoted in the Hi-Share context, and Self-Other distinction was promoted in the Lo-Share context.

An important question is whether a neural segregation of PEs is functionally meaningful. The contextual difference in cross-decoding accuracy correlated with mean $\lambda$:$\alpha$ ratio in the Hi-Share context (after controlling for accuracy and decision temperature) [Pearson correlation: $r = 0.41$, $p = 0.009$]. This behavioural measure quantifies the extent to which Self-attributed PEs contributed to learning for Other and vice versa (Fig. 4d). Thus, subjects who showed the greatest neural overlap between $PE^{self}$ and $PE^{other}$ were also those subjects that showed the strongest behavioural evidence of conflating Self- and Other-attributed learning.

**vmPFC myeloarchitecture is associated with PE adaptation.** The observed results suggest that subjects learned relations between Self- and Other-attributed computations. To probe which brain regions are important for acquiring or deploying this relational knowledge, we obtained quantitative MRI maps (Methods) of magnetisation transfer (MT), a biophysical marker of myelin density[23–25].

Social isolation causes hypomyelination in rats[26,27], which can be reversed through social re-integration[26,28]. Furthermore, false belief understanding in human infants is associated with the development of white matter tracts in so-called 'social' brain regions such as the mPFC and temporo-parietal junction[29]. On this basis, we predicted that an ability to acquire or deploy knowledge about the relations between different agents' computations would correlate with myeloarchitectural variability in these brain regions.

We conducted a whole-brain analysis to identify any regions where myelin-related MT varied, across subjects, with the context-dependent difference in cross-decoding from the fMRI analysis. Age, gender and intracranial volume were included as covariates of no interest. Here we did not measure any within-subject longitudinal structural brain changes, but correlated a measure of neural microstructure with a measure of PE reconfiguration, across subjects.

We found one significantly large cluster (Fig. 4e) of white matter adjacent to the right vmPFC [844 voxels, $p < 0.001$, whole-brain family-wise error (FWE) corrected, peak co-ordinates: $x = 12.8$, $y = 59.2$, $z = -18.4$]. Subjects with higher MT in this cluster showed a greater difference in cross-decodability between the two social contexts. This finding suggests that subjects with greater myelin density in this region may be more sensitive to learning about Self-Other relations, or deploying that relational knowledge in a context-dependent manner.

**The probability of sharing information is tracked in vmPFC.** The results presented in the previous section led us to ask what role the vmPFC might play in shaping Self-Other distinction. We hypothesised that the vmPFC might track the degree to which Self-attributed signals are associated with Other-attributed signals, using an abstract representational code, divorced from the PEs themselves. In the FBT, the probability of encountering a 'shared' trial is a proxy for the strength of this association.

We constructed a new model-based regressor (Fig. 5a) to describe subjects' perceived probability of encountering a 'shared' trial (see Methods). The model used trial-wise error-driven updates, as a function of whether a trial was 'shared' or not. The model used a single parameter, $\eta$, a learning rate that governs how quickly the subject learns about the probability of observing a 'shared' trial.

We constructed a mask of bilateral vmPFC and tested whether the blood oxygen-level dependent (BOLD) signal in this mask correlated with our new regressor (Fig. 5b). The analysis was repeated with a set of five arbitrary values of $\eta$, ranging from 0.01 to 0.1. We used Bonferroni correction to account for these repeat analyses, yielding a FWE-corrected significance threshold of $p = 0.01$. At $\eta = 0.01$, we found a significant cluster in bilateral vmPFC [192 voxels, $p = 0.008$, small-volume corrected, peak co-ordinates: $x = 3$, $y = 46$, $z = -15$]. We also conducted a whole-brain analysis (Fig. 5b). At $\eta = 0.025$ we found a significant cluster in the left lateral temporal cortex, extending to the left temporal pole [583 voxels, $p = 0.006$, whole-brain FWE-corrected, peak co-ordinates: $x = -62$, $y = -16$, $z = -16$].

These clusters may form part of a network that tracks fluctuations in the statistical relationship between Self- and Other-attributed PEs. In view of the learning rates that generated these regressors, it seems that the vmPFC may track low-frequency fluctuations whilst the lateral temporal cortex may track higher-frequency fluctuations. The mean contrast estimate within the vmPFC cluster correlated positively with the mean contrast estimate within the temporal cluster [Pearson correlation: $r = 0.45$, $p = 0.004$]. This means that subjects who showed strong evidence of tracking the low-frequency drifts in vmPFC were the same subjects who showed strong evidence of tracking the high-frequency drifts in temporal cortex.

If this relational learning process drove the PE reconfiguration that we observed, then the adaptation effect should be correlated with the relational learning effect. We found that the mean contrast estimate in the vmPFC cluster was indeed positively correlated with the adaptation effect (Fig. 5c), quantified again as the difference in Self-Other cross-decoding between the Hi-Share and Lo-Share conditions [Pearson correlation: $r = 0.35$, $p = 0.032$]. This correlation was only significant after excluding two subjects with extreme ($|Z| > 2.5$) contrast estimates. However, there was no association between mean contrast estimate in the temporal cluster and the adaptation effect, irrespective of whether outliers were excluded [Pearson correlation: $r = -0.05$, $p = 0.74$].

This is consistent with vmPFC and lateral temporal cortex tracking drifts, of different frequencies, in the statistical relationship between Self- and Other-attributed signals. The neural

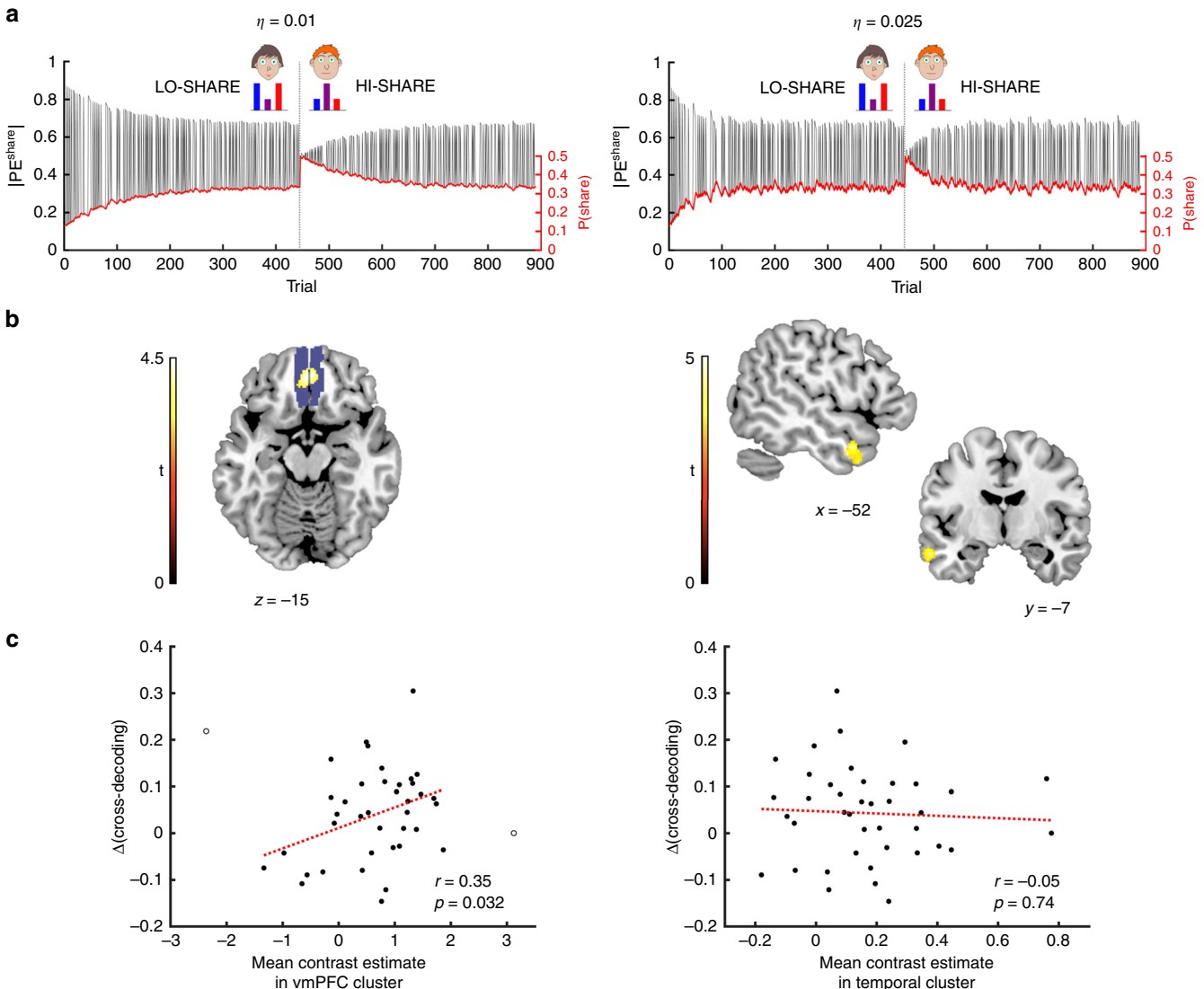

**Fig. 5 Probability of sharing information between Self and Other is tracked in vmPFC and temporal pole. a** Example regressors from a learning model that tracks the probability of encountering a 'shared' trial, P(share). The red line indicates P(share) whilst the black line indicates the PE magnitude from this model. The sharp boundary, at trial 444, indicates that this particular subject did the Lo-Share task, followed by Hi-Share. Left: Regressors generated with a learning rate ($\eta$) of 0.01. Right: Regressors generated with $\eta$ of 0.025. **b** Left: P(share), at $\eta = 0.01$, is tracked in bilateral vmPFC. Blue shading shows the mask used for small-volume correction. Right: P(share), at $\eta = 0.025$, is tracked in left temporal pole. Clusters were defined with a cluster-forming threshold of $p < 0.001$. Only clusters that were large enough to survive FWE-correction at $p < 0.01$ are displayed (additional FWE correction was applied to account for testing multiple $\eta$ values). One-sided t-contrasts on $n = 40$ independent subjects. t-statistics are shown in the colour bars. **c** Left: Mean contrast estimates in vmPFC cluster are positively correlated with the difference in Self-Other cross-decoding between the Hi- and Lo-Share conditions, after excluding two subjects with extreme contrast estimates (empty circles). Pearson correlation: $r = 0.35$ $p = 0.032$. Right: Mean contrast estimates in the temporal cluster are not correlated with the difference in Self-Other cross-decoding between the Hi- and Lo-Share conditions. Pearson correlation: $r = -0.05$ $p = 0.74$. Source data are provided as a Source Data file.

adaptation effect ought only be detectable during the test session if the statistical relationships are re-learned slowly. Conversely, fast relational learning should quickly eliminate any previously learned difference between the two contexts.

**Domain-general computation of Self-Other boundaries.** A final question is whether the neural computation of Self-Other distinction is uniquely relevant to social contexts, or whether it might also be important for attributing mental states in non-social contexts. In addition to false-belief understanding and perspective-taking, episodic thinking or 'mental time-travel' is a cognitive process that involves attributing mental states to different agents[30–32]. We used an intertemporal choice task to probe

between-subject variability in mental time-travel. We predicted that subjects who strongly distinguish between Self and Other would distinguish more between Self and future Self, and thus discount future rewards more steeply.

To test this prediction, subjects played an intertemporal choice task, choosing between immediate small monetary rewards and delayed large rewards. Choice behaviour (Fig. 6a) was well-described by a two-parameter hyperbolic discounting model (Methods and Supplementary Fig. 8). Consistent with our prediction, we found that the leak factor, $\lambda{:}\alpha$ (averaged across the Hi-Share tasks on both days, after controlling for accuracy and decision temperature) negatively correlated with the log product of the two discounting parameters [Spearman's rank

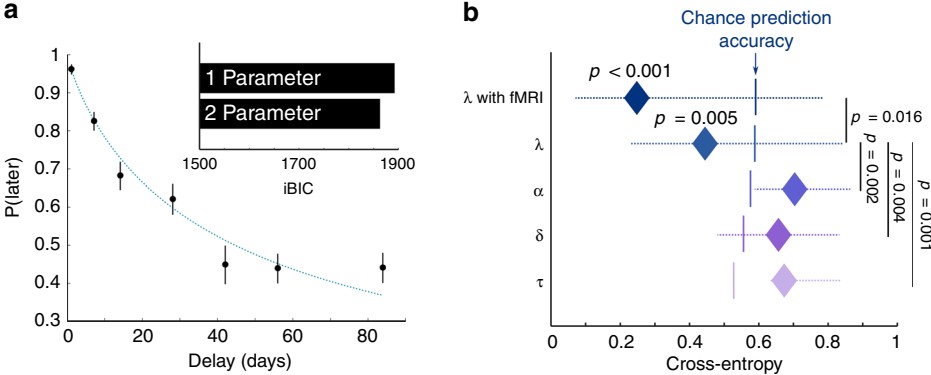

**Fig. 6 Temporal discounting propensity can be predicted by Self-Other distinction. a** Choice behaviour in the intertemporal choice task. Each scatter point shows the proportion of trials, averaged over $n = 40$ independent subjects, where the larger later option was chosen. The inset bar plot shows the relative fits of one- and two-parameter hyperbolic discounting models in terms of integrated Bayesian information criterion (iBIC). The two-parameter hyperbolic discounting curve, using the mean parameters across subjects, is shown as a blue dashed line. Error bars denote s.e.m. **b** Performance of five logistic regression models predicting subjects' temporal discounting propensities, using various predictors, with leave-one-subject-out cross-validation. Data is presented as median cross-entropy ± interquartile range, across $n = 40$ independent subjects. Lower cross-entropy denotes more accurate prediction. Discounting propensity could be predicted, above chance, using the leak parameter ($\lambda$): $p = 0.005$. Prediction accuracy was significantly better with $\lambda$ than with any other behavioural parameter from the false belief task (learning rate $p = 0.002$, memory decay $p = 0.004$, temperature $p = 0.001$). Prediction accuracy ($p < 0.001$) was significantly improved ($p = 0.016$) when Self-Other cross-decoding accuracies from the fMRI analysis were included as additional predictors in a new logistic regression model. All $p$ values were derived from two-sided permutation tests (see Methods). Vertical bars show chance level prediction accuracy for each logistic regression model. Source data are provided as a Source Data file.

correlation: $\rho = -0.32$, $p = 0.043$]. In effect, subjects who showed more Self-Other mergence displayed less discounting of future rewards.

To probe this relationship further, we attempted to classify subjects as high or low discounters using each parameter of the learning model from the FBT, using leave-one-subject-out cross-validation (Fig. 6b and Methods). Only $\lambda$ predicted discounting above chance [$p = 0.005$, permutation test]. Importantly, when fMRI measures of Self-Other mergence (cross-decodability) were included as additional training features, classification accuracy improved significantly [$p = 0.016$, permutation test]. Finally, in the same brain region where MT covaried with the fMRI training effect, we found MT was negatively associated with discount factor across subjects [$p = 0.022$, voxel-level inference, small-volume corrected] (Supplementary Fig. 9).

## Discussion

We show that the degree to which sensory PEs are expressed in agent-specific activity patterns is susceptible to experience-dependent plasticity. When tracking the beliefs of an agent with whom there had been a high proportion of shared experiences, Self-Other mergence is promoted. When tracking the beliefs of an agent with whom there had been a low proportion of shared experiences, Self-Other distinction is promoted.

Our findings show that the spatial topography of a PE signal is plastic; the way in which a learning signal is expressed in the brain can itself be learned. This adaptation may drive the learning of Self-Other boundaries that act as priors for different social contexts. For instance, we recently showed that subtle changes in how social agents are described in a cover story, are sufficient to modulate neural Self-Other distinction[20]. In light of the current findings, we predict that the degree of Self-Other PE circuit overlap should be higher between Self and a familiar other than Self and a stranger, because Self- and Other-attributed PEs will have co-occurred more in the case of the familiar other. Indeed, close interpersonal relationships have historically been described in psychology as an incorporation of Other into the Self[33]. Furthermore, neural activity patterns underpinning Self-reflection are more similar to those underpinning reflection on similar

Others than those underpinning reflection on dissimilar Others[34,35].

The FBT training was general enough to affect behaviour in an independent cognitive task. Both tasks involved the same social agents, but whilst the training task required subjects to track beliefs about numerical probabilities, the transfer task required subjects to count objects in a visual scene from different perspectives. The relational learning that occurred in FBT training was therefore not specific to learning signals, but was general enough to impact on non-learning related decision variables. The visual perspective-taking task is designed such that Self-Other distinction is required on incongruent trials but not required on congruent trials. However, we observed agent-specific transfer effects on both congruent and incongruent trials. It may be the case that when Self-Other distinction is reduced, the use of a shared model for Self and Other prevents independent sensory sampling on behalf of a single agent. Simultaneous sensory sampling on behalf of Self and Other may carry a cognitive load that slows evidence accumulation, regardless of whether the samples for Self and Other are congruent or incongruent, akin to the performance deficits seen under dual-task demands[36,37].

In localising PE signals, we found both distinct and overlapping brain regions for Self and Other, consistent with previous experiments on simulated learning[16,18]. The regions identified are consistent with previous studies examining unsigned PE signals. Extrastriate visual cortex has previously been shown to encode visual sensory surprise[38,39], whilst the intraparietal sulcus has been associated with the encoding of state prediction errors during navigation of a probabilistic environment[40]. In our main fMRI analysis we combined all of these clusters into a single multi-cluster mask, to perform multi-voxel pattern analysis. Although this analysis lacked anatomical specificity, our aim was not to localise a function, but rather to test whether representations of PEs could be changed through our training manipulation.

Consistent with our prediction, we found that myelin-related MT in ventromedial prefrontal white matter, was associated with the degree of training-induced representational change. Specifically, subjects with higher MT in this region showed a larger difference in Self-Other representational similarity between the

two conditions. This is consistent with previous research showing that mPFC microstructure in rodents[26–28,41], and macrostructure in humans[29], is a determinant of social cognition.

Our experimental design was based on the hypothesis that exposure to a strong temporal contingency between Self- and Other-attributed PEs would increase the representational similarity of these update signals through statistical learning. Conversely, we hypothesised that exposure to a weak temporal contingency between Self- and Other-attributed PEs would weaken the associative strength between these two update signals, and hence reduce their representational similarity. Our results suggested that vmPFC, along with the lateral temporal cortex, may track the associative strength between Self- and Other-attributed signals. In our study, we did not manipulate the temporal contingency between these signals per se, simply using the proportion of 'shared' trials as a proxy for this. It will be important for future work to further investigate the learning algorithm behind this adaptation of Self-Other boundaries.

The human vmPFC is involved in self-referential processing[42] and tracks the degree to which objects are associated with Self[43]. Other-attributed mental states can also be represented in the vmPFC, depending on the precise framing of the cognitive task, and the extent of Self-Other distinction[3,35,44]. We note that in our experiment, Self- and Other-attributed PEs were not themselves encoded in the vmPFC. This is not surprising because, whereas most studies on the simulation of Others' mental states have investigated value-based decision variables[3,16,44], the mental states represented in our task were visual surprise signals. The vmPFC is known to represent state and action values[45] and in studies where other agents' preferences are simulated, one should expect the contents of the mental states to be encoded in the vmPFC. In the current study, the vmPFC may be involved in mapping relations between different agents' mental states onto contextual cues, allowing them to be flexibly deployed when the relevant cue, such as a face, is re-encountered. Prior accounts describe a role for the mPFC in modulating behaviour to suit the current social context[46].

It is important to consider whether our FBT engages cognitive processes that are social per se. In essence, the task merely requires subjects to track two random variables. However, we have previously shown that the degree to which sensory PEs are encoded in agent-specific neural activity patterns depends on the social nature of the cover story[20]. Furthermore, in the current study we show that training subjects on the Hi- and Lo-Share contexts of the FBT induces a behavioural change in a visual perspective-taking task, suggesting that the FBT does tap into social processes.

Despite the above considerations, there remains a possibility that neither the FBT nor the perspective-taking task engages social cognition. The behavioural transfer from one task to the other may simply reflect a form of non-social learning[47]. Whilst we cannot say that our subjects engaged in computations that are exclusive to socially interactive settings, we nevertheless consider these computations are likely co-opted when attributing mental states to social agents. For instance, being able to represent multiple models of the same environment may be a necessary component of social cognition, whilst also useful in non-social situations. Our results show that information about agent-identity and relationships between different agents' mental states can be encoded in fundamental sensory processing signals. Whilst these signals may not be 'social' in isolation, they appear to contribute to complex social cognitive processes.

The ability to learn relationships between different agents' computations may be just one example of a form of relational learning, that is not 'social' per se. Relational learning allows organisms to represent the world efficiently. By representing

environments in terms of abstract 'concepts'[48], 'task sets'[49] or 'cognitive maps'[50], animals can rapidly generalise a structure learned in one environment to a totally new environment[51,52]. The vmPFC has also been associated with mapping latent, contextual states of the environment, in non-social situations[53,54]. Agent identity might be one example of a latent environmental state, that shapes learning and behaviour to suit the current context.

Consistent with the notion that common computations can be used in both social and non-social contexts, we found that behavioural and neural measures of Self-Other distinction are related to discounting behaviour in an intertemporal choice task. Subjects who discounted future rewards more steeply also represented other agents' mental states more distinctly from their own mental states, and were better able to distinguish the beliefs of Self and Other. This finding is consistent with a common relational learning process regulating a generalisation between Self-attributed mental states and mental states attributed to both other agents and to one's future Self. It is also consistent with prior accounts that propose a common mechanism for traversing social and temporal distances[30–32,55,56].

We considered whether a hidden variable, such as general task engagement or cognitive control, might explain the association between Self-Other distinction and temporal discounting. In intertemporal choice tasks, people with better cognitive function, across a range of tasks, tend to discount future rewards less than those with cognitive impairments[57]. We would expect this effect to promote a positive correlation between leak factor and discount factor. It is striking then that a negative association between leak factor and discount factor is detectable. We note, however, that the notion that future Self is represented like Other is, at this stage, speculative.

In summary, our results support a computational mechanism that enables updates to one model to influence change in another model. Furthermore, we show that the degree to which updates to one agent's model generalise, is itself learnable. This may facilitate generalisation of knowledge structure to new situations, and in the special case of social cognition, the generalisation from one agent's mental states to another's, enabling traversal across both social and temporal dimensions, in a flexible context-dependent fashion. The vmPFC appears to play a role in acquiring this relational knowledge, consistent with previous accounts that implicate this region in both social cognitive development[29] and abstract structure learning[53,54].

## Methods

**Participants**. 47 adults (26 female) aged 19-54, participated in a 3-day experiment. They were recruited from the UCL Institute of Cognitive Neuroscience subject pool. All participants had normal or corrected-to-normal vision and had no history of psychiatric or neurological disorders. All participants provided written informed consent, which was approved by the Research Ethics Committee at University College London, under ethics number 4446/003.

Six participants only completed days 1 and 2 of the experiment. One further participant was excluded from all analyses as it was evident, on debriefing, that they did not understand some of the tasks. This left 46 subjects (25 female) with a mean age of 26.5 (s.d. 7.8) who were included in the analysis of the visual perspective-taking task, and a subgroup of 40 subjects (22 female) with a mean age of 26.8 (s.d. 8.1) who were included in all other analyses.

**False belief task**. Subjects were trained on two probabilistic false belief tasks (FBT) on day 2, and then tested on two FBTs on day 3 (Fig. 1a). On both days, one FBT was with a male cartoon avatar and one was with a female cartoon avatar. In the FBT, subjects were instructed that they would be trying to keep track of an environment while also trying to keep track of another participant's false belief about the same environment.

The avatars represented two real participants, who had experienced what the avatars observed in a simplified version of the task. In the simplified task, participants only needed to keep track of a single fluctuating Bernoulli parameter, with no social element. Every participant played this simplified version of the task on day 1. Then on days 2 and 3, each participant was linked with trial sequences

observed on day 1 by two previous participants, one represented by the male avatar and one represented by the female avatar. This social set up was explained in a detailed, standardised way to all participants. There was no deception involved. Getting participants to play the simplified version of the task on day 1 was useful for several reasons. Firstly, it provided a stream of experiences for the next participant to model as one of the avatars. Secondly, the experience in the simplified task would make it easier for subjects to put themselves into the avatars' shoes in the FBT. Finally, the simplified task on day 1 helped participants understand how to play the FBT on days 2 and 3.

In the FBT, subjects were instructed to update their belief on 'privileged' trials, to update their estimate of the other agent's belief on 'decoy' trials, and to update both simultaneously on 'shared' trials. In the training tasks on day 2, there was a difference in the distribution of these three trial types between the two tasks. In one 'Hi-Share' context, there were 112 'privileged' trials, 112 'decoy' trials, and 224 'shared' trials. In one 'Lo-Share' context, there were 196 'privileged' trials, 196 'decoy' trials, and 56 'shared' trials. Subjects were not told about this difference in trial proportions. Each context was paired with a specific avatar. This paring was counterbalanced across subjects. In the testing tasks on day 3, there was no difference in the distribution of trial types between the two contexts. In both cases, there were 148 of each of the three trial types.

All trial sequences were pre-generated using two uncorrelated random walks. These random walks proceeded in step sizes of 0.025 and were reflected by the boundaries 0 and 1. Only pairs of walks which were uncorrelated were used to generated trial sequences. 'Privileged' samples were drawn from the first walk whilst 'decoy' samples were drawn from the second walk. 'Shared' samples were drawn randomly. We selected trial sequences which produced no correlation between trial-by-trial Self-and Other-attributed beliefs and PEs, according to behavioural simulations with a learning rate of 0.1 and no Self-Other leak.

The two social contexts were presented with two different cover stories. The mapping between social context and cover story was counterbalanced across subjects. In one cover story subjects played the role of a shop-assistant selling pink umbrellas and yellow sunshades to tourists on a tropical island, with the help of a 'shop manager' (the other agent). The fluctuating Bernoulli parameter (probability of selling a pink umbrella) was justified as changes in the weather. In the other cover story subjects played the role of a shop-assistant selling red and blue flavours of cola to tourists in a city with the help of a 'shop manager'. The fluctuating Bernoulli parameter (probability of selling a red can) was justified as changes in the popularity of the drinks, based on what was being advertised on digital displays outside the shop. Subjects were instructed that on some trials, the manager would be in a back-room and would not get to observe the sale ('privileged'). On other trials, the manager would come out of the back-room and observe the sale ('shared'). Finally, subjects were instructed that the manager was watching CCTV footage in the backroom, to track the sales, but was mistakenly watching last week's footage, so was observing misleading information ('decoy' trials). Subjects had already played in the role of the 'manager' in the simplified task on day 1 and so they were already familiar with the cover stories when they played the FBT.

On each sampling trial subjects observed a display with three visual components (Fig. 1a). At the bottom of the display subjects saw a Bernoulli outcome (e.g. pink or yellow). In the middle of the display subjects saw an image that indicated whether the trial was 'privileged', 'shared' or 'decoy'. At the top of the display subjects saw the cartoon avatar depicting the other agent whose beliefs they were trying to track. The display was presented for 1500 ms, followed by a variable inter-trial interval with a fixation cross on screen 1000–1500 ms. After 4–9 sampling trials, subjects were probed with either a Self-probe or an Other-probe. Subjects had to position an arrow along a continuous scale to report an estimate of the Bernoulli parameter, either on behalf of Self or Other. Subjects had seven seconds to give their response. They received no feedback on their performance but were told that their reimbursement at the end of the experiment depended on how well they could keep track of the environment on Self-probes and how well they could predict the choices of the other agent on Other-probes. The simplified task on day 1 presented the Bernoulli outcomes alone and subjects were only probed with Self-probe trials. Here subjects' sole task was to keep track of a single fluctuating Bernoulli parameter.

MATLAB R_2018a was used for coding the behavioural tasks, and acquiring and analysing data. Behavioural tasks were illustrated and visualised using Cogent 2000 (v125) and Cogent Graphics (v1.29).

All statistical tests performed were two-sided, unless otherwise stated. Normality in performance measures was determined with Schapiro-Wilk tests.

**Learning models.** 72 Rescorla–Wagner learning models were fit to subjects' reports on the Self-probes and Other-probes. The winning models and the estimated parameters are shown in Supplementary Fig. 1. The models tested various combinations of parameters. They all utilised parallel belief updates for Self and Other. Self-attributed PEs were modelled as 0 on 'decoy' trials. Other-attributed PEs were modelled as 0 on 'privileged' trials. This family of model has previously been shown to approximate behaviour well in the probabilistic false belief task[20]. The details of each individual model and the quality of fits are summarised in Supplementary Table 2.

The parameters included learning rate ($\alpha$), choice temperature ($\tau$), memory decay ($\delta$) and Self-Other leak ($\lambda$). These parameters could be shared between

Self-updates and Other-updates or they could be independent. The leak parameter operates like a learning rate, but updating with the wrong agent's PE. PEs were computed as the difference between the Bernoulli outcome (1 or 0) and the previous belief (of Self or Other) about the Bernoulli parameter. Beliefs were bound between 0 and 1. Behaviour is optimal in the FBT when the memory decay and leak parameters are as close to 0 as possible. Information about the optimal learning rate is provided in Supplementary Fig. 10. Supplementary Fig. 2 provides some intuition about the effects of each of these parameters on learning.

Models were fit with maximum likelihood estimation via nonlinear optimisation in MATLAB using the fmincon function. On each probe trial, the likelihood of the subject's actual response was estimated from a Beta distribution, with mode equal to the current model-derived belief, and variance equal to the temperature parameter $\tau$, fit to the individual subject. The shape parameters for this trial-specific Beta distribution were derived from this mode and variance.

The models were fit to four different datasets: The Lo-Share and Hi-Share contexts in training (day 2) and the Lo-Share and Hi-Share contexts in testing (day 3). Four model comparisons were conducted by comparing the sum of Bayesian Information Criteria (BIC) across subjects for each model. For the fMRI analysis in the test session (day 3), the same model and the same parameters were used for each subject to estimate PE regressors. The more complex of the two winning models (for Lo-Share and Hi-Share) was selected (M68). For each subject, we averaged the parameters across the two sessions, and then took the median parameter values across subjects.

Parameter recovery was measured in each of the four winning models by simulating synthetic data using each subject's fitted parameters and then re-fitting the model to the simulated data. For each parameter, we computed a between-subjects Spearman's rank correlation between generative parameter estimates and recovered parameter estimates.

Model recovery was performed by simulating choice data using each of the four best-fitting models for the four datasets, using the parameters estimated for each subject. Each of these four models was then fit to each of these four simulated datasets. For each simulated dataset, we computed the proportion of subjects for whom each of the four models was the best-fitting (lowest BIC). This gives us p(fit| sim), the probability that a model is best-fitting, given that another (or the same) model simulated the data. P(fit|sim) was converted into P(sim|fit) using Bayes' rule as follows.

$$P(sim|fit) = \frac{p(fit|sim)p(sim)}{\sum_{sim} p(fit|sim)p(sim)} \qquad (2)$$

We assumed a uniform prior on models. This technique to derive P(sim|fit) is also described in a recent review by Wilson and Collins[58].

**Perspective-taking paradigm.** This was adapted from an older paradigm where subjects had to count the number of dots in a visual scene[21]. In our adapted task, subjects had to count the number of patterns in a scene that matched a target pattern. On every trial, a target pattern was shown along with a target number. Then a room was shown with an avatar facing one wall. On every trial, there were two patterns on a wall visible to the avatar and a further two patterns on a second wall, not visible to the avatar. Some of the patterns matched the target pattern, and some of them were distractor patterns, which looked like the target pattern but were rotated 60 degrees clockwise. The orientation of the target pattern changed randomly on every trial. Subjects saw three different avatars, throughout the task. Two of these were the avatars that represented the Lo-Share and Hi-Share contexts from the FBT. The third 'avatar' was an arrow.

Each trial started with a fixation cross for 500–1000 ms, followed by the perspective that the participant was required to adopt ('YOU', 'HE', 'SHE' or 'ARROW') for 500 ms. When the cue said 'HE' or 'SHE', subjects had to adopt the perspective of the male or female avatar respectively. When the cue said 'ARROW', subjects simply had to report whether the number of target patterns that an arrow pointed to was consistent or inconsistent with a target number. Then, the target pattern and target number were displayed for 750 ms. Finally, the room, along with an avatar, was presented and subjects had up to three seconds to respond with a 'yes' or 'no' key (L and K keys on the keyboard, counterbalanced across subjects). The task consisted of 384 trials presented in a random order (Supplementary Fig. 6). Perspective (Self or Other), condition (congruent or incongruent), response (yes or no), avatar on screen (Lo-Share, Hi-Share or arrow) and avatar gaze (left or right) were balanced in a $2 \times 2 \times 2 \times 3 \times 2$ design. Subjects played the same visual perspective-taking task on day 1, before FBT training, and then again on day 2, after FBT training.

**Drift–diffusion modelling.** A drift–diffusion model was fit to the visual perspective-taking task using the fast-dm-30.2 toolbox[59] with maximum likelihood estimation. All trials where subjects responded too slowly (>3000 ms) or quickly (<500 ms) were excluded from the analysis. 1.4% of trials were excluded in total.

Correct responses were faster than incorrect responses at both baseline [paired t-test: t(45) = −10.35, p < 0.001] and transfer [paired t-test: t(45) = −7.93, p < 0.001]. In order to allow the model to generate different response time distributions for correct and incorrect responses, we allowed for between-trial drift rate variability. This was achieved by randomly sampling the drift rate on each trial from one of twelve possible Gaussian distributions. These distributions accounted for different

trial types, namely the three avatars (Hi-Share, Lo-Share, Arrow), two perspectives (Self, Other) and two conditions (congruent, incongruent). These twelve distributions had separate mean parameters but shared a subject-specific variance parameter. All twelve of these mean drift rate parameters were shared for 'yes' and 'no' responses. Additional parameters included non-decision time (mean and variance), drift starting point (mean and variance) and boundary separation distance (mean and variance). These three parameters were randomly sampled on each trial from Gaussian distributions with mean and variance fit to each subject's dataset. Unlike the drift rate, the distributions for these three parameters did not vary as a function of trial type. The model was fit twice to each subject, once for the baseline dataset (day 1) and again for the transfer dataset (day 2).

The arrow trials were included to obtain a measure of repeat training effects that were not related to the FBT. By subtracting any changes in performance on arrow trials from changes in performance on the Hi-Share and Lo-Share trials, we could quantify training effects over and above those that were merely due to repeat exposure to the visual perspective-taking task. For each subject, for each of the twelve drift rate parameters, we subtracted the estimated parameter at baseline (day 1) from the estimated parameter at transfer (day 2) to obtain a change score. The change scores were then corrected by subtracting from them the change scores for arrow trials. A positive corrected change score indicated that the relevant drift rate increased more than the increase seen on arrow trials. A negative corrected change score indicated that the relevant drift rate increased less than the increase seen on arrow trials. Normality of parameter estimates was determined with Schapiro-Wilk tests.

**MRI data acquisition**. Scanning took place in a 3T whole-body MRI scanner (Magnetom Prisma system from Siemens Healthcare, Erlangen, Germany) with a body coil for transmission and a 64-channel receive head coil. We collected the functional data with four 2D EPI scanning sessions (two runs for the Lo-Share context and two runs for the Hi-Share context). Each volume comprised 40 slices with a resolution of 3 mm isotropic, with a TR of 2.8 s, TE of 30 ms, slice tilt of −30°, and Z-shim of −0.4. Heart rate was monitored using a Nonin 8600FO pulse-oximeter and respiration rate was monitored using a Siemens breathing belt during scanning. Following the functional scans, a field mapping sequence was used to measure inhomogeneity of the B0 field. This was a double-echo fast low-angle shot (FLASH) sequence with a short TE of 10 ms and a long TE of 12.46 ms.

Lastly, a multiple parameter mapping protocol was applied for microstructural imaging[60]. Three 3D multi-echo FLASH acquisitions were made, with predominantly T1, proton density (PD) and magnetisation transfer (MT) weighting respectively. The flip angle was 6° for the PD-weighted and MT-weighted images, and 21° for the T1-weighted images. MT-weighting was achieved through the application of a Gaussian radio-frequency pulse 2 kHz off-resonance with 4 ms duration and a nominal flip angle of 220°. The data were acquired with whole-brain coverage at an isotropic resolution of 0.8 mm. Gradient echoes were acquired with alternating readout gradient polarity at eight equidistant echo times ranging from 2.3 to 18.4 ms in steps of 2.3 ms. Only six echoes were acquired for the MT-weighted acquisition in order to maintain a TR of 25 ms of all volumes.

Prior to each FLASH acquisition, two additional low resolution (8 mm isotropic) volumes were acquired, one with the 64-channel head and neck array coil and the other with the body coil. A single echo, with a TE of 2.2 ms, was acquired in each case using a 6° flip angle and a TR of 6 ms. The acquisition time of each of these calibration volumes was 5.9 s. These 'sensitivity maps' were used to correct the position-specific modulation of the receive sensitivity field.

**fMRI pre-processing**. The first six volumes of each functional run were discarded. Slice-timing correction was applied. Motion correction was carried out using the 'realign and unwarp' toolbox within SPM12. Images were co-registered to the first volume acquired for each subject. The motion-corrected images were then unwarped using the field map. The functional images were co-registered to the respective subject's MT map, normalised into Montreal Neurological Institute (MNI) space and then smoothed with a Gaussian kernel of full-width at half maximum (FWHM) 8 mm isotropic. Physiological data were converted into 18 nuisance regressors with the PhysIO Toolbox v7.2.0[61].

**PE localisation**. Two separate general linear models (GLM) were estimated, one for localising PE[self] and one for localising PE[other]. These variables were not correlated with each other (Supplementary Fig. 11). The Self-GLM modelled the onsets of 'privileged' and 'shared' trials, parametrically modulated by |PE[self]|. The Other-GLM modelled the onsets of 'shared' and 'decoy' trials, parametrically modulated by |PE[other]|. We used unsigned PE signals because we were not interested in the component of the signal that corresponded to the stimulus outcome (colour), but rather the surprise component of the signal. Temporal and dispersion derivatives were also included. All regressors were z-scored within subjects. The onsets of probes were included in both GLMs, as were 24 nuisance regressors, describing motion and physiological noise. First-level maps were entered into a one-sided t-test at the second level. Significantly large clusters ($p < 0.05$ FWE-corrected) were identified at the group-level using a cluster-forming threshold of $p < 0.001$, in a whole-brain analysis, using Gaussian random field theory (GFRT) to control the FWE rate.

We then conducted a searchlight multi-voxel pattern analysis using The Decoding Toolbox (version 3.994)[62]. We obtained trial-specific activation patterns by fitting a different GLM for each trial to normalised but unsmoothed images. One regressor represented the onset of the trial of interest, and one regressor represented the onsets of all other trials. Temporal and dispersion derivatives of these two regressors were also included. 24 nuisance regressors, describing motion and physiological noise, were included. A beta map was produced that represented the contribution of the trial of interest to the whole functional run. Decoding analyses were performed on these beta maps. We ran two whole-brain searchlight procedures, one for |PE[self]| and one for |PE[other]|. For each analysis we trained a least absolute shrinkage and selection operator (LASSO) linear regression model on three runs of functional data, to predict |PE| from the BOLD signal across voxels within spherical searchlights of radius four voxels.

The regression model was then tested on the fourth, held-out run of functional data. Performance was quantified as the Fisher Z-transformed correlation between the model's predicted |PE| values and the actual |PE| values. The transformed correlation coefficients for each of the four test sets were averaged to produce a mean cross-validated decoding accuracy, which was attributed to the voxel at the centre of the searchlight. This was repeated for each subject with a range of L1 penalty parameters ($10^{-5}$ to $10^{-3}$ in increments of $2.5 \times 10^{-5}$). The whole-brain accuracy maps were then smoothed with a Gaussian kernel of FWHM 8 mm isotropic.

We extracted decoding accuracies from voxels masked by the co-ordinates of significantly large clusters ($p < 0.05$, whole-brain FWE-corrected for cluster-extent) identified in the two respective mass-univariate GLM analyses. For each subject, an optimal penalty parameter was selected for the PE[self] analysis, by identifying which penalty produced the highest median decoding accuracy across masked voxels in the PE[other] analysis. Concurrently, optimal penalty parameters were selected for the PE[other] analysis by identifying which penalties produced the highest median decoding accuracies across masked voxels in the PE[self] analysis. By optimising the hyperparameters for one analysis on a different analysis, we mitigated the risk of overfitting.

Once penalty parameters were selected for each subject, first-level decoding accuracy maps were entered into a one-sided t-test at the second level. Significantly large clusters ($p < 0.05$ FWE-corrected) were identified at the group-level using a cluster-forming threshold of $p < 0.001$, in a whole-brain analysis, using GFRT to control the family-wise error rate. These clusters from the PE[self] and PE[other] searchlight analyses were combined to form a single Self-Other multi-cluster mask. This mask was used to select voxels for the subsequent decoding analyses.

**Self-Other classification analysis**. Per subject, we randomly sampled pairs of trials on either side of the median unsigned PE magnitude and subtracted the BOLD signal in the low PE-magnitude trial from signal in high PE-magnitude trial. This was performed first for PE[self] (excluding 'decoy' trials) and then again for PE[other] (excluding 'privileged' trials). This resulted in a series of contrast images ('pseudotrials') that each comprised a noisy representation of |PE[self]| and another series of contrast images that each comprised a noisy representation of |PE[other]|.

We used a two-step decoding approach. The first step was a feature extraction step, using principal components analysis (PCA). This reduced the dimensionality down from approximately 10,000 voxels (see PE localisation in Methods) to approximately 100 components. The second step involved training a LASSO logistic regression model on these components to classify pseudotrials as being PE[self] or PE[other].

This approach required tuning of two hyperparameters, the L1 penalty and the percentage variance-explained by the principal components. We used nested cross-validation to optimise these two hyperparameters. We used a grid-search, sampling over a range of L1 values ($10^{-5}$ to $10^{-3}$ in increments of $2.5 \times 10^{-5}$) and a range of variance-explained percentages (90%, 92.5%, 95%, 97.5%). Two pseudotrials from each class were randomly sampled to constitute a hold-out set. The remainder constituted a training set. For each possible pair of hyperparameter values, 40 inner folds of cross-validation were performed, by randomly sampling two pseudotrials of each class from the training set. Hyperparameters were selected that produced the lowest median cross-entropy across folds.

$$\text{Cross Entropy} = -[y\ln(p) + (1-y)\ln(1-p)] \qquad (3)$$

Here, $y$ denotes the true binary class label (Self or Other) and $p$ denotes the probability of the pseudotrial being PE[self], assigned by the classifier. Therefore, cross-entropy is lowest when confident accurate predictions are made, and it is highest when confident inaccurate predictions are made.

Finally, the classifier with optimised hyperparameters was applied to the hold-out set, and cross-entropy was measured. This whole procedure was repeated for 40 outer folds of cross-validation and performance was quantified as the median cross-entropy across the 40 outer folds. This analysis was conducted twice, once for the Hi-Share context and once for the Lo-Share context.

**Cross-decoding analysis**. This analysis followed the same pipeline as described for the Self-Other classification analysis, but instead of training a logistic regression model, a linear regression model was trained with labels of PE[self] magnitudes and tested to predict PE[other] magnitudes, and vice versa. Here, there were only eight outer folds and one inner-fold of cross-validation, due to the natural split in train

and test sets (see Supplementary Fig. 12). For tuning hyperparameters, the Fisher Z-transformed correlation between predicted PE magnitudes and true PE labels was maximised. Final performance was quantified as the mean Fisher Z-transformed correlation across the eight outer folds of cross-validation. This analysis was conducted twice, once for the Hi-Share context and once for the Lo-Share context.

**MT analysis.** Quantitative MT maps were created and then spatially processed using the hMRI toolbox v0.2.0 in SPM12[63]. Spatial processing involved three steps: segmentation, diffeomorphic deformation and tissue-weighted smoothing. Each map was converted into grey matter (GM), white matter (WM) and cerebrospinal fluid (CSF) tissue class images. Tissue class images were iteratively aligned from all of the subjects to their own average before normalising the images to MNI space. Finally, tissue-weighted smoothing was performed with a Gaussian kernel of FWHM 6 mm isotropic. The resulting maps only included those voxels with an a priori probability of being considered in the relevant tissue class (GM, WM or CSF) above 5% and an original tissue density above 5%.

We conducted a whole-brain, mass-univariate regression at the group level. The fMRI cross-decoding effect [Hi-Share – Lo-Share] was the independent variable, and white matter MT was the dependent variable within each voxel. Age, gender and intracranial volume were included as covariates. Significantly large clusters ($p < 0.05$ FWE-corrected) were identified using a cluster-forming threshold of $p < 0.001$, in a whole-brain analysis, using GFRT to control the family-wise error rate.

To ensure our analysis had adequate statistical power to detect interindividual variability in MT maps, we looked at effect sizes from previous studies that conducted regression analyses on MT maps. Allen et al. reported a peak voxel effect of $r = 0.62$ with a sample size of 48 participants, when correlating metacognitive ability with MT in a whole-brain analysis[64]. Steiger et al. reported a peak voxel effect of $r = 0.69$ with a sample size of 31 participants, when correlating memory ability with MT in a whole-brain analysis[65]. To detect an effect size (correlation) of 0.64 with a type I error rate of $\alpha = 0.001$ (cluster-forming threshold) and a type II error rate of $\beta = 0.1$ (90% power), we would need a sample size of 39 participants[66]. As our sample size in this analysis was 40 participants, we consider we are adequately powered to detect effects of similar magnitude to those seen in previous studies of interindividual MT variability.

**Analysis of shared trial tracking.** We constructed a Rescorla–Wagner model to track the likelihood of observing a 'shared' trial:

$$P_{t+1} = P_t + \eta(S - P_t) \tag{4}$$

Where $P_t$ is the probability of observing a 'shared' trial on trial $t$, $\eta$ is a learning rate parameter and $S$ indicates whether the trial was shared, coded as 1 on 'shared' trials and 0 on 'privileged' and 'decoy' trials. We assumed that subjects had learned, on day 2, that the probability of observing a 'shared' trial was 12.5% in the Lo-Share condition and 50% in the Hi-Share condition, providing starting estimates for the two conditions during the fMRI session on day 3.

For each subject, we constructed a single GLM to explain BOLD signal variability using the $P$ regressor along with its temporal and dispersion derivatives. The regressor was z-scored prior to estimating the GLM. We repeated the analysis five times, using a different value of $\eta$ each time. These values were 0.01, 0.025, 0.05, 0.075 and 0.1. We first ran a region of interest (ROI) analysis, using a mask of bilateral vmPFC, hand-defined as part of the Automated Anatomical Labelling ROI library[67]. We then ran a whole-brain analysis. Significantly large clusters ($P < 0.05$ FWE-corrected) were identified using a cluster-forming threshold of $p < 0.001$, using GFRT to control the FWE rate for multiple comparisons across voxels and then applied additional Bonferroni correction to control the FWE rate for multiple comparisons across the five values of $\eta$.

In this analysis, the model-based regressor was z-scored within each subject, before estimating GLMs and each subject's BOLD activation. It has recently been shown that inter-individual variability in BOLD activation is highly sensitive to the way that regressors are pre-processed, particularly to whether they are z-scored or not[68]. We found that the results of this analysis were not sensitive to whether or not the $P$ regressor was z-scored prior to estimating the GLM.

**Intertemporal choice task.** On day 1, before any other tasks, participants answered sixty binary forced-choice questions. Each question presented the subject with a choice between an immediate small reward and a delayed larger reward. Immediate rewards were monetary gifts ranging from £1 to £9, which would be given to the participant at the end of the experiment. Delayed rewards were monetary gifts ranging from £2 to £10, which would be given to the participant after some temporal delay. The set of possible delays comprised: one day, one week, two weeks, four weeks, six weeks, eight weeks and twelve weeks. All subjects were presented with the same sixty questions, selected pseudo-randomly to ensure that all magnitudes and delays were sampled. Subjects were instructed to choose consistently with their subjective preferences. They were told that one of their sixty choices would be selected at random at the end of experiment, and they would receive the chosen monetary amount after the chosen temporal delay.

**Delay discounting models.** We fit two discounting models to subjects' inter-temporal choice behaviour. The first model was a hyperbolic discounting model:

$$V_{\text{later}} = \frac{V}{1 + kD} \tag{5}$$

$V_{\text{later}}$ denotes the value of the later option after passing it through the discount function. $V$ denotes the raw, undiscounted value of the option. $D$ denotes the delay of the reward, in days, and $k$ is a free parameter. The second model was a two-parameter hyperbolic discounting model, where $D$ is exponentiated by an additional discounting parameter $S$, representing Stevens' power law time perception[69]:

$$V_{\text{later}} = \frac{V}{1 + kD^S} \tag{6}$$

In both models, the difference between the values of the two options was passed through a softmax function to account for probabilistic choice behaviour.

$$P_{\text{later}} = \frac{1}{1 + e^{\beta(V_{\text{sooner}} - V_{\text{later}})}} \tag{7}$$

$P_{\text{later}}$ denotes the probability of the subject choosing the 'larger later' option on a specific trial. $\beta$ is a free parameter that governs choice stochasticity.

Models were fit with unconstrained parameter values in log space. Thus, in native space, parameter values had a lower bound of zero and no upper bound. We optimised the maximum a posteriori (MAP) of observed data, given a likelihood and an empirical group level prior over model parameters. The hyperparameters (mean and variance) of this Gaussian prior were estimated by maximising the likelihood of all the data from all subjects. To optimise the hyperparameters of the prior distribution we used an expectation-maximisation (E-M) algorithm, that iterates between E-steps, where posterior parameter distributions are estimated for each subject, and M-steps, where the empirical prior is updated. The algorithm iterates between these two steps until convergence.

At every E-step, a MAP estimate is computed for each subject by minimising the negative log posterior probabilities with the fminunc function in MATLAB. The variance on this MAP parameter is computed using a Laplace approximation, which assumes that the posterior distribution is simply a Gaussian around the MAP estimate.

These subject-specific means and variances are then used for updating the hyperparameters of the prior on the M-step. Here, the mean and variance of the Gaussian prior are updated. The mean is simply set to the mean of all subjects' MAP estimates. The variance update incorporates both subject-level estimation error and between-subject variability:

$$\text{Prior variance} = \frac{1}{N} \sum_j^N \left[ \left(m_j - \mu_j\right)^2 + \sigma_j^2 \right] \tag{8}$$

The variance of the prior distribution incorporates the individual variance ($\sigma^2$) of the posterior for each subject $j$, as well as the deviation of each subject's mean ($m$) from the prior mean ($\mu$). $N$ denotes the total number of subjects. To compare the goodness-of-fit for each model, we computed an 'integrated BIC' score (iBIC) for each model, approximating the evidence for the full hierarchical model at the group level[70].

Since both parameters in the two-parameter model contribute to discounting, we defined an aggregate measure of discounting as $\log(kS)$ and validated this as a reasonable measure of discounting propensity by testing its correlation with a model-free measure of discounting behaviour, area under the discounting curve (Supplementary Fig. 8). We conducted a leave-one-subject-out logistic regression analysis to examine if we could predict whether a subject was a high or low discounter, via a median split on $\log(kS)$, using parameters from the learning models, fit to FBT data. Model performance was quantified as the median cross-entropy across cross-validation folds, where $y$ was the true binary label (high or low discounter) and $p$ was the probability of being a high discounter that the classifier assigned to the subject.

Statistical inference was made by generating permutation-based null distributions. For each statistical test, the analysis was simulated 5000 times, each time randomly permuting the class labels. This generated a null distribution to derive $p$ values, describing whether the prediction accuracy was significantly better than chance. This technique also allowed us to quantify whether the prediction accuracy achieved using one parameter was significantly better or worse than the prediction accuracy achieved using other parameters. To make these statistical inferences we constructed null distributions of cross-entropy differences.

**Unanalysed data.** Subjects did not take part in any other behavioural tasks as part of this experiment but they did fill out personality questionnaires on days 1 and 2 of the experiment, to assess for subclinical personality traits. These included the Beck Depression Inventory, Community Assessment of Psychic Experience, Interpersonal Reactivity Index, Empathy Quotient, Inventory of Callous-Unemotional traits, the Borderline Scale of the Personality Assessment Inventory and the Borderline Personality Questionnaire.

**Reporting summary.** Further information on research design is available in the Nature Research Reporting Summary linked to this article.

## Data availability

All data generated during this study are freely available on the Open Science Framework at [osf.io/62mza/]. The source data underlying Figs. 2a, b, 3b, c, 4b–d, 5c, 6a, b, Supplementary Figs. 1, 3a, b, 6b, c and 8 and Supplementary Tables 1 and 2 are also provided as a Source Data file. A reporting summary for this Article is available as a Supplementary Information file.

## Code availability

Custom MATLAB code for behavioural modelling is freely available on the Open Science Framework at [osf.io/62mza/].

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

## Acknowledgements

This work was supported by the Wellcome Trust. R.J.D. receives an Investigator Award (098362/Z/12/Z) and the Wellcome Centre for Human Neuroimaging receives core centre funding (091593/Z/10/Z). The Max Planck UCL Centre is a joint initiative supported by UCL and the Max Planck Society. S.E. conducted this work as a pre-doctoral fellow of the International Max Planck Research School on Computational Methods in Psychiatry and Ageing Research (IMPRS COMP2PSYCH). The participating institutions are the Max Planck Institute for Human Development, Berlin, Germany, and University College London, London, UK. For more information, see: [https://www.mps-ucl-centre.mpg.de/en/comp2psych]. T.U.H. is supported by a Wellcome Sir Henry Dale Fellowship (211155/Z/18/Z), a grant from the Jacobs Foundation (2017-1261-04), the Medical Research Foundation, and a 2018 NARSAD Young Investigator grant (27023) from the Brain & Behavior Research Foundation.

## Author contributions

S.E. and Z.K-N. conceived the project. S.E. designed the experiment, collected the data, analysed the data and wrote the initial draft of the paper. T.U.H. advised on fMRI data analysis, provided code and edited the paper. R.M. advised on the drift-diffusion modelling and edited the paper. G.W.S. provided code for fitting discounting models to intertemporal choice behaviour and edited the paper. R.J.D. and Z.K-N. supervised the project and edited the paper. R.J.D. acquired funding for the project.

## Competing interests

The authors declare no competing interests.
