## [Peer Review File · Nature Communications]

Reviewers' comments:

Reviewer #1 (Remarks to the Author):

Ereira et al. present an investigation of prediction errors in shaping the self-other distinction. Participants in their experiment engaged in a probabilistic false belief task in which they attended to predict a binary outcome, while also predicting what another person would predict. The task manipulated self-other overlap by varying the degree of shared vs. nonshared information presented to the participant other target person. Behavioral results in the task and a transfer task (visual perspective taking) as well as computational modeling, fMRI classification, and analysis of white matter myelination all supported the hypothesis that shared prediction errors increased the overlap between self and other. In general, I found this paper to be a thorough, detailed, and rigorous exploration of how shared experience shifts the boundary between self and other. I think the questions asked and answered in this paper would be a broad interest to social psychologists/neuroscientists. The authors have also made a considerable effort to make their complex methods accessible to these communities. I think this paper could potentially be a valuable addition to the literature, but there are a few issues which I believe it would benefit from addressing:

1) The false belief task that the authors use is not terribly naturalistic. It effectively consists of social window dressing on a probabilistic learning task. I took a quick look at the additional details on this task presented in the earlier investigation which the authors cite, and that window dressing some seem to be high quality, for what it is. However, how confident can we be that this task tapped into social processes per se? As I understand it, since there was no deception, participants never believed that there was a real person experiencing what the avatar saw. In principle a participant could have performed well by ignoring the nominally social content and just focusing on underlying logic of the task. If one repeated the experiment without framing it as a false belief task with multiple agents, etc., how confident could we be that the results would differ? And in what way would they be expected to do so?

2) Related to (1), I find it somewhat concerning that fMRI analyses do not seem to implicate a particularly canonical set of social brain regions. I realize that these analyses are attempting to localize prediction error signals rather than social information per se. However, it would be reassuring – and help address (1) – if the authors could show (even in supplementary materials) that they replicate some standard effects like the self>other univariate effect in vMPFC. Similarly, it would be informative to try a whole brain (e.g., searchlight or parcel-based) analog of the self-other classification analysis they currently perform using only the PE regions.

3) How did the authors control the family-wise error rate in the MRI analyses? They state a cluster forming threshold, but not the algorithm. Given that they say they are using SPM12, I imagine that the default SPM Gaussian random field correction (GFT) was used? This method has been shown fairly compellingly to be too liberal (e.g., Eklund et al., 2016). As such I think it would be prudent to re-run these corrections with an algorithm more robust to violations of assumptions, such as maximal statistic permutation testing (implemented, for instance, in SnPM). In any case, the authors should state explicitly how FWE control was achieved.

4) The authors place considerable emphasis on the term plasticity, but I worry that this may be somewhat overselling their results. Although I suppose plasticity can be used to refer to changes in behavior or representations, it is much more typically used to describe changes in the brain – particularly, in brain structure (white and grey matter). The authors do measure white matter and find an interesting association between myelination in vmPFC and change in functional MRI self-other overlap. However, as the authors themselves point out, there is no way of knowing from these analyses whether there was any direct cause and effect between the training and vmPFC myelination, or its direction. In other words, there is no direct evidence that the structure of the brain has change as a result of the training. To my mind any large changes of this kind would be exceedingly surprising, given the brevity of the false belief task. The authors do not claim otherwise, but I am concerned that using words like ‘plasticity’ and ‘rewiring’ in the context of a training/learning task and a white matter measure might create a misleading impression on causal or less technical readers. I think the authors should scale back their use of these terms – particularly in title and abstract – and take more steps to avoid this misapprehension where these terms still appear.

5) A couple of minor issues: a) I know that Nature group publications require a statement to the effect that “All statistical tests performed were two sided.” However, I do not believe they require it repeated in each section of the methods. b) Also, while the authors do a good job of describing their more complex analyses, they sometimes do not specify what test they are conducting in some of the simpler cases (e.g. t-tests). Since t-statistics are common to many analyses in the linear model family, I found myself guessing in some cases precisely which test was conducted.

Reviewer #2 (Remarks to the Author):

The authors examined whether the Self-Other distinction is susceptible to experience-dependent plasticity by training subjects in a mentalizing task. They concomitantly assessed delay discounting and perspective taking. They also acquired neural data in the form of both fMRI and structural MRI. They found that a higher percentage of shared trials reduces the self-other boundaries, that this effect percolates the perspective taking task and that it is correlated to delay discounting. At the neural level they observe a lower cross decoding for the low-share compared to the high-share PEs and the Δ cross-decoding is correlated to myelin in the vmPFC.

Conceptual

One of the claims of the paper (in the authors’ own words) is that “the cognitive Self-Other boundaries develop through simple learning processes”. However, they do not really demonstrate that. They show that there is some learning process involved and they propose one model, but in the absence of a model comparison involving different (i.e., “non-simple”) learning processes, we are presented with no actual evidence about the exact computations of of Self-Other boundaries development. In other terms: what is the alternative computational hypothesis?

The neural results

The paper presents two main neural results: i) the self-other boundaries (as proxied by the capacity of cross-decoding PE in the self vs. other condition) decrease as a function of sharing the same experience; ii) the change in cross decoding is correlated by a structural change (myelin) in the vmPFC cortex. Something is clearly missing in this model, as the PE correlated are not in the vmPFC. What is the link between the PE correlated in the parietal cortex and the vmPFC plasticity? In the present form, the neural findings look like a series of facts, without a unifying theory.

Dependent variable definition

There is an issue concerning the choice of the computational dependent variable. For example, the cross classification scores is correlated with the ratio between the lambda and the alpha, while the discount factor is correlated with the lambda alone. The authors should decide which version of the computational parameter (lambda alone or lambda/alpha) is the more relevant for them and stick to it in the whole paper. Otherwise one might legitimately be tempted to think that they are fishing for significant correlations, by adjusting the dependent variable.

Interpretation of the delay - discounting task.

The authors administered an additional task to evaluate reward discounting and found a significant correlation between discount and self-other separation. Interpret this correlation as supporting the idea that steep delay discount depends on a strong distinction between the self and the future self. This interpretation is, at best, far fetched. First, there is no consensus on the fact that delay discounting actually relies on a representation of the future self (as a different person). Second, there is at least another much simpler interpretation, that is that subjects that were less engaged in the experiment put less effort in both tasks (i.e., exerting less cognitive control).

Interpretation of the parameters

The computational results are hard to interpret as we do not know what are the optimal values of the model parameters (which, especially in the case of learning rates, is very task dependent). Also model simulations with different parameters values would help understand their function.

Modeling

The authors administered three tasks (the main prediction task, the perspective taking task and the) and analysed the three tasks with different modeling frameworks: associative learning, DDM and hyperbolic discounting. No effort is made (except in some cases correlations) to integrate the three tasks into a unified framework. On a less theoretical note, could the authors reduce the number of parameters in their models by showing that some of them are the same across tasks?

Minor concerns:

Manuscript ergonomics: put the equations (at least the one concerning the belief update) in the result section.

Data visualization: please show the individual points and the values of the correlations of the recovery analysis.

Lines 41-46: this part of the introduction seems to be off the topic, as the study does not investigate signed reward prediction errors, but rather unsigned perceptual prediction errors.

Line 13: "Humans achieve this by simulating each other's computations in agent-specific neural circuits. What". This is the second line of the abstract, but I don't think there is enough consensus on this topic to include this sentence in the "background" of the abstract.

Line 14: "To test this," To test what? No clear hypothesis is formulated in the previous sentence.

Line 16: "Long-lasting". Really? Can an effect lasting one day be considered "long lasting"?

Reviewer #3 (Remarks to the Author):

In this paper, Ereira and colleagues test participants on a previously reported false-belief-task (FBT) that requires participants to update their knowledge about fluctuating probabilities for themselves and for another person separately. The task comprises different trial types that enable self-updates, other-updates or both-updates. Participants performance in this task can be modified by pre-training them on different versions of the task that differ in the relative amount of both-update trials (shared trials). People perform a schedule with fewer both-update trials better compared to one with more shared trials and this translates also to subsequent performance on balanced schedules and also has some links to a visual perspective taking task and a temporal discounting task. Modelling results suggest that increased numbers of shared trials lead to increased correlations of self-other beliefs in the subsequent balanced task. Neural correlates with unsigned prediction errors exist and decoding analyses suggest activity in these regions differ based on the previous schedule. The key claim of the paper is that agent-specific neural PE circuits adapt to social context; increased shared experiences lead to a long-lasting increased overlap of self- and other-attributed neural PE circuits.

I very much appreciate the effort and manpower that went into conducting the study. Testing was performed on several days, with a large subject pool, with and without MRI, on different tasks and the battery of analyses methods is equally impressive. It is striking that task behaviour on the equivalent balanced schedules on the final day differs as a function of the previous training and that beliefs about self and other seem more aligned during that session if there had been more shared trials in the past. The neural results are in my view the weakest part of the paper. Given the claim about agent-specific neural circuits, the reader expects to learn something about where these circuits are located in the brain; but the study does not arrive at a conclusion about a single brain region or a meaningful network of regions. The correlations of the FBT task performance with other tasks are interesting, but they also cast doubt on the specificity of the behavioural and neural effects found in the FBT task.

Major points:

1. Neural results: Usually the advantage of MRI is that it allows identification of specific brain regions. This study is unusual in that the main analyses of interest merges activity patterns over several functionally and anatomically distinct regions. This would not be so bad if the regions identified in the first place formed a meaningful network, but this is not the case. In this form, the conclusion about merged/separated PE patterns does not go much beyond what we already know from the authors previous MEG paper. Neither are these regions anywhere near (except maybe the supramarginal gyrus) any of the many brain regions usually identified in social cognition studies. I am surprised that their neural key regressor, the absolute prediction error ($\text{abs}(\text{PE})$) does not identify other regions, given that is very much linked to expected payoff (participants are paid for holding accurate beliefs and $\text{abs}(\text{PE})$ indicates the degree of holding an (in)accurate belief). The study holds several potentially interesting variables that would allow a more precise description of the role of the brain regions and this should be exploited more. I am a bit worried though, that the tight temporal spacing does not provide enough jitters for a fully parametric event-related design. Was this also the reason why $\text{abs}(\text{PE-S})$ and $\text{abs}(\text{PE-O})$ were not put in the same GLM although they are reasonably decorrelated? The MT vmPFC result is puzzling and does not relate to the other results in a meaningful way.

2. The relationships between the tasks is interesting. However, they also cast doubt on the specificity of the initial finding about plasticity of learning about self and other belief updates. The perspective taking task suggests the result is not specific to learning and the temporal discounting result suggests it is not specific to social cognition. This is particularly a concern because the identified regions, too, are not very much related to either social cognition or learning. It raises the possibility that the investigated phenomenon is much less specific than the lambda parameter (indicating self-other misattribution of PEs) suggests.

3. Behavioural modelling: Along the lines of the previous point – the lambda parameter is perhaps the most important parameter in the model because it relates to the degree that PEs are mis-assigned to the wrong agent which sets the rationale for the PE merging MRI decoding. However, each of the 4 sessions has a different best fitting model casting doubt on whether the lambda indeed the critical variable. Can the authors show that:

- The differences in correlated beliefs (Fig 2b) do not emerge from models without the lambda (using model simulations and the next-best fitting models that don't contain a lambda)?
- Analogous to the parameter recovery in Fig. 2d, can model recovery based on model simulations using a confusion matrix reliably identify their winning models? Given that overall 72 models were fitted to the data, this seems to be the more pertinent control compared to the parameter recovery analysis.
- The lambda effects persist when controlling for overall performance and decision noise?

Minor points:

1. I am sceptical whether the visual perspective taking result really maps onto the FBT experiment. The

authors make the point in the discussion that “ we observed agent-specific transfer effects on both congruent and incongruent trials.” However, supplementary Fig. 6 looks as if RT and accuracy benefits are mostly in the congruent condition, which does not map very well on the FBT results. I understand that the conclusions are derived from the drift diffusion model that modelled several drift rates for different conditions relative to the control condition. However, I could not work out a) which ones of those are shown in Fig.3c and b) whether panel 3c indeed shows evidence for separate drift rate effects relating to both congruent and incongruent trials.

2. Task performance is measured by the correlation of participants predictions with the true Bernoulli parameter. While there is some plausibility to this measure as an index of performance, it ignores some aspects of the predictions like the range or the intercept and it also deviates from the authors own measure of performance in previous paper on this paradigm. Why was this measure chosen and do the results in fig.2a hold for other measures of performance such as the one used previously or the one implicit in the RL model?

3. The fMRI regressors are derived from the more complex RL model which fits only one of the two MRI sessions best. Why was this model chosen – rather than for example the model that fits best when averaging over both of the sessions? That seems to allow for a fairer comparison between the session types. Could the differences in MRI self-other decoding be related to this modelling choice?

4. Typo page 3 line 54: “recircuit”

5. Personally, I don't find the summary figure very useful or intuitive. This comment is not about the science behind the paper and I would leave it up to the authors if they want to change it or not. I would guess that changing that figure can help convey the author's findings to a broader audience.

Blue = reviewer comments

Black = response to reviewer

Reviewer 1

Ereira et al. present an investigation of prediction errors in shaping the self-other distinction. Participants in their experiment engaged in a probabilistic false belief task in which they attempted to predict a binary outcome, while also predicting what another person would predict. The task manipulated self-other overlap by varying the degree of shared vs. nonshared information presented to the participant other target person. Behavioral results in the task and a transfer task (visual perspective taking) as well as computational modeling, fMRI classification, and analysis of white matter myelination all supported the hypothesis that shared prediction errors increased the overlap between self and other. In general, I found this paper to be a thorough, detailed, and rigorous exploration of how shared experience shifts the boundary between self and other. I think the questions asked and answered in this paper would be of broad interest to social psychologists/neuroscientists. The authors have also made a considerable effort to make their complex methods accessible to these communities. I think this paper could potentially be a valuable addition to the literature, but there are a few issues which I believe it would benefit from addressing.

Thanks for the kind feedback and constructive comments. We hope that the reviewer finds our response helpful, and that the suggested revisions make for a stronger article.

1) The false belief task that the authors use is not terribly naturalistic. It effectively consists of social window dressing on a probabilistic learning task. I took a quick look at the additional details on this task presented in the earlier investigation which the authors cite, and that window dressing seems to be high quality, for what it is. However, how confident can we be that this task tapped into social processes per se? As I understand it, since there was no deception, participants never believed that there was a real person experiencing what the avatar saw. In principle a participant could have performed well by ignoring the nominally social content and just focusing on underlying logic of the task. If one repeated the experiment without framing it as a false belief task with multiple agents, etc., how confident could we be that the results would differ? And in what way would they be expected to do so?

Firstly, we would like to clarify that there was indeed a real person experiencing what the avatar saw. In this sense, the task is truly “social”. We have added the following text to clarify:

Lines 77-78: “The other player was a real participant, playing a simplified version of the game (Methods).”

Lines 581-592: “The avatars represented two real participants, who had experienced what the avatars observed in a simplified version of the task. In the simplified task, participants only needed to keep track of a single fluctuating Bernoulli parameter, with no social element. Every participant played this simplified version of the task on day 1. Then on days 2 and 3, each participant was linked with trial sequences observed on day 1 by two previous participants, one represented by the male avatar and one represented by the female avatar. This social set up was explained in a detailed, standardised way to all participants. There was no deception involved. Getting participants to play the simplified version of the task on day 1 was useful for several reasons. Firstly, it provided a stream of experiences for the next participant to model as one of the avatars. Secondly, the experience of the simplified task would make it easier for subjects to put themselves into the avatars’ shoes in the FBT. Finally, the simplified task on day 1 helped participants understand how to play the FBT on days 2 and 3.”

Lines 630-632: “The simplified task on day 1 presented the Bernoulli outcomes alone and subjects were only probed with Self-probe trials. Here subjects’ sole task was to keep track of a single fluctuating Bernoulli parameter.”

In our previous study (Ereira et al. 2018), by comparing the task with a non-social control condition, we showed that the social framing of the task is important for determining how the PEs are represented in the brain. However, the reviewer is correct that a participant could perform well by ignoring the social content. In fact, an important finding of the paper is an association between Self-Other distinction on the false belief task, and intertemporal reasoning on a delay discounting task (Fig. 6). We hypothesised that the social task taps into domain-general computations that are not exclusively reserved for social cognition, but are necessary for the development of Self-Other boundaries. In our revisions we have added substantial discussion of these issues:

Lines 494-541: It is important to consider whether our FBT engages cognitive processes that are social per se. The task does not emulate a natural social environment; there is no back-and-forth dyadic interaction, nor do subjects observe the behaviour of other agents. In essence, the task merely requires subjects to track two random variables, and perhaps the minimally social nature of the task has no bearing on the cognitive processes at play.

However, we have previously provided evidence indicating subjects are indeed sensitive to the social context of the task. We showed previously that the degree to which sensory PEs are encoded in agent-specific neural activity patterns depends on the social nature of the cover story²³. In this prior experiment, we compared the social task with a non-social task, where for the latter there was no other actual participant. Instead, subjects imagined themselves in a counterfactual situation in which they were the one exploiting the corrupted stream of information. The social and non-social tasks were structurally identical, differing only in their cover stories and this manipulation alone was sufficient to modulate the extent to which Self- and Other-attributed PEs were neurally distinct. Furthermore, in the current study we show that training subjects on the Hi- and Lo-Share contexts of the FBT induces behavioural change in a visual perspective-taking task, suggestive that the FBT does tap into social processes.

Despite the above considerations there remains a possibility that neither the FBT nor the perspective-taking task engages social cognition. Thus, a behavioural transfer from one task to the other may simply reflect a form of non-social learning⁵³ that might be evident even if alternative versions of the tasks were conducted, for instance using arrows with different visual features as opposed to avatars with different faces. It is difficult to assess what makes a task ‘social’. Whilst we cannot say with absolute certainty that our subjects engaged in computations exclusive to socially interactive settings, we nevertheless consider these computations are likely to be co-opted when attributing mental states to social agents. For instance, being able to represent multiple models of the same environment may be a necessary component of social cognition, whilst also useful in non-social situations. Our results show that information about agent-identity and relationships between different agents’ mental states can be encoded in fundamental sensory processing signals. Whilst these signals in isolation are not ‘social’ per se, they appear to contribute to complex social cognitive processes.

The ability to learn relationships between different agents’ computations may be just one example of a form of relational learning, that is not social per se. Relational learning allows organisms to represent the world efficiently. By representing environments in terms of abstract ‘concepts’⁵⁴, ‘task sets’⁵⁵ or ‘cognitive maps’⁵⁶, animals can rapidly generalise a structure learned in one environment to a totally new environment^{57,58}. The vmPFC has also been associated with mapping latent, contextual states of the environment, in non-social situations^{59,60}. Agent identity may simply be one example of a latent environmental state, that shapes learning and behaviour to suit the current context.

Consistent with the notion that common computations can be used in both social and non-social contexts, we found that behavioural and neural measures of Self-Other distinction are related to discounting behaviour in an intertemporal choice task. Subjects who discounted future rewards more steeply also represented other agents’ mental states more distinctly from their own mental states, and were better able to distinguish the beliefs of Self and Other. This finding is consistent with a common relational learning process regulating a generalisation between Self-attributed mental states and mental states attributed to both other agents and to one’s future Self. It is also consistent with prior theoretical accounts that propose a common mechanism for traversing social and temporal distances^{33-35, 61}.

2) Related to (1), I find it somewhat concerning that fMRI analyses do not seem to implicate a particularly canonical set of social brain regions. I realize that these analyses are attempting to localize prediction error signals rather than social information per se. However, it would be reassuring – and help address (1) – if the authors could show (even in supplementary materials) that they replicate some standard effects like the self>other univariate effect in vmPFC. Similarly, it would be informative to try a whole brain (e.g., searchlight or parcel-based) analog of the self-other classification analysis they currently perform using only the PE regions.

As the reviewer notes, our main fMRI analyses are of sensory prediction errors rather than a social versus non-social contrast. However, we agree with the suggestion to do a sanity check along the lines suggested. We now add a new analysis where we contrast Self probe trials with Other probe trials. This Other > Self contrast yielded a cluster in the posterior region of the right temporoparietal junction, consistent with previous work on Theory of Mind. This is now supplementary Fig. 13 (see below).

The Self > Other contrast did not yield any activation in the vmPFC. We have, however, conducted a new analysis, in response to comments from reviewer #2. This analysis shows that the vmPFC and temporal pole both track the probability of experiencing a 'shared' trial. We have incorporated these findings into a new main text figure (Fig. 5). Please see our response to reviewer #2 and the manuscript for further details on these new findings.

3) How did the authors control the family-wise error rate in the MRI analyses? They state a cluster forming threshold, but not the algorithm. Given that they say they are using SPM12, I imagine that the default SPM Gaussian random field correction (GFT) was used? This method has been shown fairly compellingly to be too liberal (e.g., Eklund et al., 2016). As such I think it would be prudent to re-run these corrections with an algorithm more robust to violations of assumptions, such as maximal statistic permutation testing (implemented, for instance, in SnPM). In any case, the authors should state explicitly how FWE control was achieved.

We agree that we were unclear in the manuscript about how we controlled the family-wise error rate and we thank the reviewer for picking this up. Indeed, we used Gaussian random-field theory (GRFT). Eklund et al. showed that clusterwise inference with GRFT can lead to false positives if a liberal cluster-forming threshold (e.g. $p < 0.01$) is used. At a liberal threshold, large clusters are formed, and over these large clusters, the actual spatial autocorrelation function (SACF) deviates from the Gaussian assumption. However, at a more conservative cluster-forming threshold (e.g. $p < 0.001$), the clusters have a smaller radius and the SACF is well approximated by a Gaussian.

In Eklund et al. 2018 (Human Brain Mapping), the authors state that “For cluster inference, where groups of voxels are tested together by looking at the size of each cluster, we found that parametric methods perform well for a high cluster defining threshold (CDT; $p = .001$) but result in inflated false positive rates for low CDTs (e.g., $p = .01$)”

In our original analyses we used the advised CDT of $p < 0.001$. We have now made our Methods section clearer about the precise correction technique used:

Lines 793-795: Significantly large clusters ($P_{FWE} < 0.05$) were identified at the group-level using a cluster-forming threshold of $P < 0.001$, in a whole-brain analysis, using Gaussian random field theory (GRFT) to control the family-wise error rate.

In light of the reviewer’s concern, we also re-ran our clusterwise corrections using maximal statistic permutation testing with SnPM (10,000 permutations). We found that all of the identified clusters were still significantly large at $P_{FWE} < 0.05$. Given that the parametric clusterwise method with $CDT < 0.001$ is generally considered valid and is also common practice, we left the corrected p-values as they were in the manuscript.

4) The authors place considerable emphasis on the term plasticity, but I worry that this may be somewhat overselling their results. Although I suppose plasticity can be used to refer to changes in behavior or representations, it is much more typically used to describe changes in the brain – particularly, in brain structure (white and grey matter). The authors do measure white matter and find an interesting association between myelination in vMPFC and change in functional MRI self-other overlap. However, as the authors themselves point out, there is no way of knowing from these analyses whether there was any direct cause and effect between the training and vMPFC myelination, or its direction. In other words, there is no direct evidence that the structure of the brain has change as a result of the training. To my mind any large changes of this kind would be exceedingly surprising, given the brevity of the false belief task. The authors do not claim otherwise, but I am concerned that using words like ‘plasticity’ and ‘rewiring’ in the context of a training/learning task and a white matter measure might create a misleading impression on causal or less technical readers. I think the authors should scale back their use of these terms – particularly in title and abstract – and take more steps to avoid this misapprehension where these terms still appear.

We thank the reviewer for raising this important point. Since we found behavioral and representational changes lasting 24 hours, it is likely that synaptic plasticity is implicated. However, we acknowledge that we don't directly measure this synaptic plasticity, and that as such, the language used in the paper might have been misleading in this regard, especially since we are not describing structural brain changes. To address this, we have made the following edits in the title, abstract and text:

Title: Social training ~~rewires~~ reconfigures prediction errors to shape Self-Other boundaries.

Lines 48-49: a high-level learning process ought to enable the brain to ~~recircuit~~ reconfigure low-level learning signals...

Line 18-19 (abstract): ~~This plasticity adaptation was associated with intersubject myeloarchitectural variability in ventromedial prefrontal white matter. Subjects with higher magnetisation transfer in ventromedial prefrontal white matter, a marker of myelin density, exhibited stronger adaptation.~~

Line 215 (results subtitle): ~~Plasticity~~ Adaptation of neural Self-Other distinction

Line 280 (results subtitle): Ventromedial prefrontal myeloarchitecture is associated with PE ~~plasticity~~ reconfiguration

Lines 422-424 (discussion): This ~~plasticity~~ representational adaptation may drive the learning of Self-Other boundaries that act as priors for different social contexts.

Lines 458-460 (discussion): Consistent with our prediction, we found that myelin-related MT in ventromedial prefrontal white matter, was associated with the degree of training-induced ~~plasticity~~ representational change.

We completely agree with the reviewer that it is unlikely that our training procedure substantially altered myelination in the vmPFC. Our interpretation is that individuals with greater vmPFC myelination are more prone to training-induced changes in PE representations. We have added text to make this clearer:

Line 293-298: We conducted a whole brain analysis to identify any regions where myelin-related MT varied, across subjects, with the context-dependent difference in cross-decoding from the fMRI analysis. Age, gender and intracranial volume were included as covariates of no interest. Readers should note that, in this analysis, we did not measure any within-subject longitudinal structural changes. Rather, we correlated a measure of brain structure with a measure of representational change, across subjects.

5) A couple of minor issues:

a) I know that Nature group publications require a statement to the effect that “All statistical tests performed were two sided.” However, I do not believe they require it repeated in each section of the methods.

To improve readability, we have deleted all instances of this statement, except for the first, which we have kept as follows:

Lines 632-633: All statistical tests performed were two-sided, unless otherwise stated.

b) Also, while the authors do a good job of describing their more complex analyses, they sometimes do not specify what test they are conducting in some of the simpler cases (e.g. t-tests). Since t-statistics are common to many analyses in the linear model family, I found myself guessing in some cases precisely which test was conducted.

We have revised the manuscript so that every time a statistical result is presented, the statistical test conducted is stated in brackets.

Reviewer 2

The authors examined whether the Self-Other distinction is susceptible to experience-dependent plasticity by training subjects in a mentalizing task. They concomitantly assessed delay discounting and perspective taking. They also acquired neural data in the form of both fMRI and structural MRI. They found that a higher percentage of shared trials reduces the self-other boundaries, that this effect percolates the perspective taking task and that it is correlated to delay discounting. At the neural level they observe a lower cross decoding for the low-share compared to the high-share PEs and the Δ cross-decoding is correlated to myelin in the vmPFC.

We thank the reviewer for providing insightful comments. We have addressed each comment and suggestion, and trust our revisions make for a stronger manuscript.

Conceptual

One of the claims of the paper (in the authors' own words) is that "the cognitive Self-Other boundaries develop through simple learning processes". However, they do not really demonstrate that. They show that there is some learning process involved and they propose one model, but in the absence of a model comparison involving different (i.e., "non-simple") learning processes, we are presented with no actual evidence about the exact computations of of Self-Other boundaries development. In other terms: what is the alternative computational hypothesis?

We agree with the reviewer, and accept our phrasing here was imprecise. The referee is correct in stating we don't explicitly assess what kind of learning mechanism is involved in forming Self-Other boundaries. We show that a Self-Other mergence is promoted by sharing a high proportion of experiences with Other, and that a Self-Other distinction is promoted by simulating another's experiences that are not shared with Self. The effects of training last at least a day, generalize to other tasks, and are associated with a change in the representation of prediction errors.

We have changed the abstract:

Lines 21-22 (abstract): Our findings suggest that Self-Other boundaries ~~develop through simple learning processes~~ are learnable variables, shaped by the statistical structure of social experience.

We have also conducted an entirely new analysis that we believe goes some way to assess what kind of learning process might underlie our results. We describe this in more detail in response to the reviewer's next comment.

The neural results: The paper presents two main neural results: i) the self-other boundaries (as proxied by the capacity of cross-decoding PE in the self vs. other condition) decrease as a function of sharing the same experience; ii) the change in cross-decoding is correlated by a structural change (myelin) in the vmPFC cortex. Something is clearly missing in this model, as the PE correlations are not in the vmPFC. What is the link between the PE correlations in the parietal cortex and the vmPFC plasticity? In the present form, the neural findings look like a series of facts, without a unifying theory.

This is an astute comment – and it has inspired us to perform an entirely new analysis.

In the original submission, we found that subjects with higher myelin-related MT in the vmPFC had greater PE adaptation, and we therefore speculated that the vmPFC may track the associative strength between Self- and Other-attributed belief updates.

Inspired by the referee we now test this idea directly. Thus, we performed a new fMRI analysis using a model-derived estimate of the probability of seeing a shared trial. We found that this signal is encoded in vmPFC. One interpretation of this is that the vmPFC is tracking the relationship between Self-attributed experiences and Other-attributed experiences, with downstream effects on the coding of prediction errors.

We have made a new Figure 5 in the main text with these results:

Lines 306-353: The results presented in the previous section led us to ask what role the vmPFC might play in shaping Self-Other distinction, given that the PEs were not encoded in this region. We hypothesised that the vmPFC might track the degree to which Self-attributed signals are associated with Other-attributed signals, using an abstract representational code, divorced from the PEs themselves. In the FBT, the probability of encountering a shared trial can be considered a proxy for the strength of this association.

We constructed a new model-based regressor (Fig. 5a) to describe subjects' perceived probability of encountering a shared trial. We assume that subjects had learned perfectly, on day 2, that this probability was 12.5% in the Lo-Share condition and 50% in the Hi-Share condition, providing different priors for the two conditions during the fMRI session on day 3. The model used trial-wise error-driven updates, as a function of whether a trial was shared or unshared (privileged or decoy). The model used a single parameter, η , a learning rate that governs how quickly the subject learns about the likelihood of observing a shared trial.

We constructed a mask of bilateral vmPFC and tested whether the BOLD signal in this mask correlated with our new regressor (Fig. 5b). The analysis was repeated with a set of five values of η , ranging from 0.01 to 0.1. We used Bonferroni correction to account for these repeated analyses, yielding a corrected significance threshold of $P_{FWE} = 0.01$. At $\eta = 0.01$, we found a significant cluster in bilateral vmPFC [192 voxels, $P_{FWE} = 0.008$, small-volume corrected for cluster extent, peak co-ordinates: $x = 3, y = 46, z = -15$]. We also conducted a whole-brain analysis (Fig. 5b). At $\eta = 0.025$ we found a significant cluster in the left lateral temporal cortex, extending to the left temporal pole [583 voxels, $P_{FWE} = 0.006$, cluster level, peak co-ordinates: $x = -62, y = -16, z = -16$].

These clusters may form part of a network that tracks fluctuations in the statistical relationship between Self- and Other-attributed signals. In view of the learning rates that generated these regressors, it seems that the vmPFC may track low-frequency fluctuations whilst the lateral temporal cortex may track higher-frequency fluctuations. The mean contrast estimate within the vmPFC cluster correlated positively with the mean contrast estimate within the temporal cluster [Pearson correlation: $r = 0.45$, $P = 0.004$], indicating subjects who showed strong evidence of tracking the low-frequency drifts in vmPFC were the same subjects who showed strong evidence of tracking the high-frequency drifts in temporal cortex.

If this relational learning process was driving the sensory PE reconfiguration that we observed, then the adaptation effect should be correlated with the relational learning effect. We found that the mean contrast estimate in the vmPFC cluster was indeed positively correlated with the adaptation effect (Fig. 5c), quantified again as the difference in Self-Other cross-decoding between the Hi-Share and Lo-Share conditions [Pearson correlation: $r = 0.35$, $P = 0.032$]. This correlation was only significant after excluding two subjects with extreme ($|Z| > 2.5$) contrast estimates. However, there was no association between mean contrast estimate in the temporal cluster and the adaptation effect, irrespective of whether outliers were excluded [Pearson correlation: $r = -0.05$, $P = 0.74$].

The above findings are consistent vmPFC and lateral temporal cortex tracking drifts, of different frequencies, in the statistical relationship between Self- and Other-attributed signals. The context-dependent difference in Self-Other PE overlap ought only be detectable if the relationship between Self- and Other-attributed signals is re-learned slowly. This should allow the two different Self-Other relationships, learned on the previous day, to continue to impact the PE representations. Conversely, fast relational learning, should quickly eliminate any difference between the two social contexts, as it drives subjects to quickly learn that there is no difference between the two contexts during the fMRI session.

Fig. 5 | Probability of sharing information between Self and Other tracked in vmPFC and temporal pole. *a*, Example regressors from a learning model that tracks the probability of encountering a shared trial, $P(\text{share})$. The red line indicates $P(\text{share})$ whilst the black line indicates the unsigned PE signal from this model. The sharp boundary, at trial 444, indicates that this particular subject did the Lo-Share task, followed by Hi-Share. Left: Regressors generated with a learning rate (η) of 0.01. Right: Regressors generated with η of 0.025. *b*, Left: $P(\text{share})$, at $\eta = 0.01$, is tracked in bilateral vmPFC. Blue shading shows mask used for small-volume correction. Right: $P(\text{share})$, a $\eta = 0.025$, is tracked in left temporal pole. Clusters were defined with a cluster-forming threshold of $P < 0.001$. Only clusters that were large enough to survive correction at $P_{\text{FWE}} < 0.01$ are displayed (additional FWE correction was applied to account for testing multiple values). *c*, Left: Mean contrast estimates in vmPFC cluster are positively correlated with the difference in Self-Other cross-decoding between the Hi- and Lo-Share conditions, after excluding two subjects with extreme contrast estimates. These two subjects are visualised with the two empty circles. Right: Mean contrast estimates in the temporal cluster are not correlated with the difference in Self-Other cross-decoding between the Hi- and Lo-Share conditions.

We have added further details of this analysis to the Methods section:

Lines 893-905: We constructed a model to track the likelihood of observing a shared trial:

$$P(\text{share})_{t+1} = P(\text{share})_t + \eta(\text{outcome} - P(\text{share})_t)$$

Where $P(\text{share})$ is the probability of observing a shared trial on trial t , η is a free learning rate parameter and outcome is coded as 1 on shared trials and 0 on privileged and decoy trials. For each subject, we constructed a single GLM to explain BOLD signal variability using the P regressor along with its temporal and dispersion derivatives. We repeated the analysis five times, using a different value of η each time. These values were 0.01, 0.025, 0.05, 0.075 and 0.1. We first ran a region of interest (ROI) analysis, using a mask of bilateral vmPFC, hand-defined as part of the Automated Anatomical Labelling ROI library⁶⁷. We then ran a whole-brain analysis. Significantly large clusters ($P_{\text{FWE}} < 0.05$) were identified using a cluster-forming threshold of $P < 0.001$, in a whole brain analysis, using GFRT to control the family-wise error rate for multiple comparisons across voxels and then additional Bonferroni correction to control the family-wise error rate for multiple comparisons across learning rates.

We have also added a paragraph to the discussion, describing more clearly the motivation behind the study design, as well as the limitations of our conclusions:

Lines 467-479 (discussion): Our experimental design was based on the hypothesis that exposure to a strong temporal contingency between Self- and Other-attributed PEs would increase the representational similarity of these update signals through statistical learning. Conversely, we hypothesised that exposure to a weak temporal contingency between Self- and Other-attributed PEs would weaken the associative strength between these two update signals, and hence reduce their representational similarity. Our results suggested that vmPFC, along with the lateral temporal cortex, two brain regions connected via the uncinate fasciculus⁴⁷, may track the degree to which Self- and Other-attributed signals are associated with each other. In our study, we did not manipulate the temporal contingency between Self- and Other-attributed signals per se, simply using the proportion of shared trials as a proxy for this. It will be important for future work to investigate further the learning algorithm behind the adaptation of Self-Other boundaries, and what features of information sharing are necessary and sufficient for this adaptation.

Dependent variable definition: There is an issue concerning the choice of the computational dependent variable. For example, the cross classification scores is correlated with the ratio between the lambda and the alpha, while the discount factor is correlated with the lambda alone. The authors should decide which version of the computational parameter (lambda alone or lambda/alpha) is the more relevant for them and stick to it in the whole paper. Otherwise one might legitimately be tempted to think that they are fishing for significant correlations, by adjusting the dependent variable.

We understand what the reviewer is saying. We had reasoned that each quantity was logically motivated by the specific analysis. The lambda:alpha ratio is the most meaningful measure of Self-Other mergence, but for assessing relationships with the delay discounting task we wanted to ensure that the association was specific to the lambda parameter. However, we agree as to the importance of avoiding both actual, and the appearance of, fishing.

The lambda:alpha ratio and the raw lambda values are highly correlated ($r = 0.94$, $p < 0.0001$), and as recommended by the reviewer, we now use the lambda:alpha ratio for both correlation analyses:

Lines 384-387: Consistent with our prediction, we found that the leak factor, $\lambda:\alpha$ (averaged across the Hi-Share tasks on both days, after controlling for accuracy and decision temperature) negatively correlated with the log product of the two discounting parameters [Spearman's rank correlation: $\rho = -0.32$, $P = 0.043$].

Interpretation of the delay - discounting task. The authors administered an additional task to evaluate reward discounting and found a significant correlation between discount and self-other separation. Interpret this correlation as supporting the idea that steep delay discount depends on a strong distinction between the self and the future self. This interpretation is, at best, far fetched. First, there is no consensus on the fact that delay discounting actually relies on a representation of the future self (as a different person). Second, there is at least another much simpler interpretation, that is that subjects that were less engaged in the experiment put less effort in both tasks (i.e., exerting less cognitive control).

We agree that this idea is speculative. Although there is a substantial literature linking discounting to psychological distance from future self (Ainslie 1989, Buckner and Carroll 2007, Trope and Liberman 2010, Soutchek et al. 2016, Hill et al. 2017), our observation of a correlation between Self-Other separation and discount factor is only a single data point. Although we find the idea fascinating, we agree with the referee and have changed the text (see below) to emphasize that much more work is needed to reach a strong conclusion on this observation.

However, we disagree that disengagement would be likely to produce the observed correlation. We would expect a disengaged subject to exhibit high Self-Other mergence (failure to distinguish Self and Other) and high discounting (high impulsivity, low cognitive

control). General disengagement should therefore produce a positive association between these variables. The fact that we find a negative association is therefore particularly interesting. We have added a new paragraph to the discussion, to expand on these ideas:

Lines 542-551: We considered whether a hidden variable, such as general task engagement or cognitive control, might explain the association between Self-Other distinction and temporal discounting. In intertemporal choice tasks, people with better cognitive function, across a range of tasks, tend to discount future rewards less than those with cognitive impairments^{62,63}. We would expect this effect to promote a positive association between leak factor and discount factor. It is striking that a negative association between leak factor and discount factor is detectable, in spite of the opposing effect that would be driven by general task engagement and cognitive control. However, the notion that future Self is represented like Other is, at this stage, speculative. Our finding is only suggestive and future work will be needed to explore a relationship between interpersonal and intertemporal reasoning.

Interpretation of the parameters - The computational results are hard to interpret as we do not know what are the optimal values of the model parameters (which, especially in the case of learning rates, is very task dependent). Also model simulations with different parameters values would help understand their function.

Thank you for the suggestion. We have added an analysis of optimal model parameters, and a set of simulations showing the effects of different parameter values:

Lines 657-660: Behaviour is optimal in the FBT when the memory decay and leak parameters are as close to 0 as possible. Information about the optimal learning rate is provided in supplementary Fig. 10. Supplementary Fig. 2 provides some intuition about the effects of each of these parameters on learning.

Supplementary Fig. 10 | Optimal learning rate for the FBT. Simulations were used to assess the relationship between task performance on the FBT and learning rate. The optimal learning rate is approximately 0.07.

Supplementary Fig. 2 | The functions of the learning parameters. Simulations were used to generate plots to illustrate the effect of varying α, δ and λ in the FBT. a, The left plot shows the belief trajectories (attributed to Self and Other) for an example subject with a low learning rate (α). The right plot shows the equivalent with a high α . Increasing α makes the belief trajectories more volatile. b, The left plot shows belief trajectories for an example subject with a low memory decay (δ). The right plot shows the equivalent with a high δ . Increasing δ reduces the step size taken along the belief trajectories on each trial, keeping them closer to the starting value of 0.5. c, The left plot shows belief trajectories for an example subject with a low leak parameter (λ). The right plot shows the equivalent with a high λ (shared between the two update equations). Increasing λ shifts both belief trajectories so that they are more correlated in time.

Modeling

The authors administrated three tasks (the main prediction task, the perspective taking task and the) and analysed the three tasks with different modeling frameworks: associative learning, DDM and hyperbolic discounting. No effort is made (except in some cases correlations) to integrate the three tasks into a unified framework. On a less theoretical note, could the authors reduce the number of parameters in their models by showing that some of them are the same across tasks?

The reviewer is right that we used different modelling frameworks for each of the three behavioural tasks. This was motivated by the fact the tasks themselves are quite different from each other, and each is associated with a set of well-established analysis techniques in the literature. Our approach was therefore to analyze each task with its own best-understood computational principles, and then to relate the tasks to each other using the quantities derived from these analyses.

It is an interesting idea to try to create a generalised model, unifying the three tasks. We think this is outside the scope of the current work. An additional concern is that this approach might introduce spurious correlations between the tasks, which we have tried to avoid.

Minor concerns: Manuscript ergonomoy: put the equations (at least the one concerning the belief update) in the result section.

We have added the belief update equations to the results section.

Data visualization: please show the individual points and the values of the correlations of the recovery analysis.

We have amended the figures as requested,

Lines 41-46: this part of the introduction seems to be off the topic, as the study does not investigate signed reward prediction errors, but rather unsigned perceptual prediction errors.

Our motivation for including this was that most of the work on simulated (social) prediction errors has been in the reward domain, and we wanted to call out the relevant prior work.

However, we can see that it was a bit too much, so we have now significantly shortened this section.

Line 13: “Humans achieve this by simulating each other’s computations in agent-specific neural circuits. What”. This is the second line of the abstract, but I don’t think there is enough consensus on this topic to include this sentence in the “background” of the abstract.

We thank the reviewer for pointing out this statement, which is indeed too strong for the background section of the abstract. We have now changed this sentence:

“Recent evidence suggests humans might achieve this by simulating each other’s computations in agent-specific neural circuits...”

Line 14: “To test this,” To test what? No clear hypothesis is formulated in the previous sentence.

Thanks for catching this. We have rephrased as follows:

“...but it is not known how circuits might become agent-specific. Here we investigated whether agent-specificity adapts to social context. We trained subjects...”

Line 16: “Long-lasting”. Really? Can an effect lasting one day be considered “long lasting”?

We have made this more specific to make sure this is not misleading for readers:

“Training altered the agent-specificity of prediction error (PE) circuits for at least 24 hours...”

Reviewer 3

In this paper, Ereira and colleagues test participants on a previously reported false-belief-task (FBT) that requires participants to update their knowledge about fluctuating probabilities for themselves and for another person separately. The task comprises different trial types that enable self-updates, other-updates or both-updates. Participants performance in this task can be modified by pre-training them on different versions of the task that differ in the relative amount of both-update trials (shared trials). People perform a schedule with fewer both-update trials better compared to one with more shared trials and this translates also to subsequent performance on balanced schedules and also has some links to a visual perspective taking task and a temporal discounting task. Modelling results suggest that increased numbers of shared trials lead to increased correlations of self-other beliefs in the subsequent balanced task. Neural correlates with unsigned prediction errors exist and decoding analyses suggest activity in these regions differ based on the previous schedule. The key claim of the paper is that agent-specific neural PE circuits adapt to social context; increased shared experiences lead to a long-lasting increased overlap of self- and other-attributed neural PE circuits.

I very much appreciate the effort and manpower that went into conducting the study. Testing was performed on several days, with a large subject pool, with and without MRI, on different tasks and the battery of analyses methods is equally impressive. It is striking that task behaviour on the equivalent balanced schedules on the final day differs as a function of the previous training and that beliefs about self and other seem more aligned during that session if there had been more shared trials in the past. The neural results are in my view the weakest part of the paper. Given the claim about agent-specific neural circuits, the reader expects to learn something about where these circuits are located in the brain; but the study does not arrive at a conclusion about a single brain region or a meaningful network of regions. The correlations of the FBT task performance with other tasks are interesting, but they also cast doubt on the specificity of the behavioural and neural effects found in the FBT task.

We thank the reviewer for these comments. The constructive feedback was very helpful, and we hope have addressed the reviewer's concerns.

Major points: 1. Neural results: Usually the advantage of MRI is that it allows identification of specific brain regions. This study is unusual in that the main analyses of interest merges activity patterns over several functionally and anatomically distinct regions. This would not be so bad if the regions identified in the first place formed a meaningful network, but this is not the case.

It is true that our main fMRI analysis is unconventional in that we sacrificed anatomical specificity to achieve a sensitive discrimination of prediction error representations. However, unlike many other studies using fMRI, our aim was not to localise any particular cognitive function, but rather to test whether representations of PEs can change as a

function of a training manipulation. We have added to the discussion so that this is more explicit:

Lines 454-457: In our main fMRI analysis we combined all of these clusters into a single multi-cluster, to perform multi-voxel pattern analysis. Although this analysis lacked anatomical specificity, our aim was not to localise a function, but rather to test whether representations of PEs could be changed through our training manipulation.

In this form, the conclusion about merged/separated PE patterns does not go much beyond what we already know from the authors previous MEG paper.

We disagree and consider our current results extend significantly our previous findings. Our previous paper showed that Self- and Other-attributed PEs are neurally distinct. The central idea behind the current manuscript was to perform an *intervention*, manipulating the degree to which Self and Other share information. Our key, and novel, finding is that this manipulation alters the degree to which Self- and Other-attributed PEs are neurally distinct. This was not testable with our previous dataset.

Neither are these regions anywhere near (except maybe the supramarginal gyrus) any of the many brain regions usually identified in social cognition studies.

As noted in the reply to reviewer #1, our main fMRI analyses were concerned with sensory prediction errors (rather than a social versus non-social contrast), and so we wouldn't necessarily expect to identify 'social' brain regions per se. However, in response to both reviewers' comments, we have added a new analysis where we contrast Other probe trials against Self probe trials. This contrast -- which is a reasonable proxy for 'social vs non-social' -- identifies the right TPJ, a region that is widely considered to form part of the 'Theory of Mind' network. Whilst this finding doesn't really extend the main results of our paper, it is a reassuring 'sanity check'. The results are described in the new supplementary figure 13. (Also see response to reviewer #1.)

I am surprised that their neural key regressor, the absolute prediction error (abs(PE)) does not identify other regions, given that is very much linked to expected payoff (participants are paid for holding accurate beliefs and abs(PE) indicates the degree of holding an (in)accurate belief).

The reviewer is absolutely right. The reason why no reward-related regions were identified is because we used a positive contrast, whereas it is $-abs(PE)$ that is associated with payoff (albeit fairly weakly). If we run a negative contrast on PE (collapsing of self- and Other-PE in a single GLM) we do in fact see bilateral caudate activity. We haven't included this result in the paper, since we feel it is not directly pertinent to our hypotheses and other results, but would be happy to include it if the reviewer thinks it is necessary.

The study holds several potentially interesting variables that would allow a more precise description of the role of the brain regions and this should be exploited more. I am a bit worried though, that the tight temporal spacing does not provide enough jitters for a fully parametric event-related design. Was this also the reason why $\text{abs}(\text{PE-S})$ and $\text{abs}(\text{PE-O})$ were not put in the same GLM although they are reasonably decorrelated?

We determined the task design by simulating GLMs with many possible designs, and choosing one that achieved a high task-efficiency, which also contained enough trials to fit the behavioural models. Although the task has tight temporal spacing we are still able to estimate the parameters of the GLM. We chose to estimate $\text{abs}(\text{PE-S})$ and $\text{abs}(\text{PE-O})$ in separate GLMs to remain consistent with how we conducted this analysis in our previous paper. We are, however, able to exploit our parametric event-related design in a single GLM. Indeed, when identifying brain regions for $-(\text{PE})$, which identified bilateral caudate, and for the Self>Other contrast, which identified right TPJ, we used a single GLM. Furthermore, we used a single GLM for a new analysis, which is described in response to the reviewer's next point. We believe that this new analysis makes good use of our parametric design, localising some interesting task-related variables to specific brain regions.

The MT vmPFC result is puzzling and does not relate to the other results in a meaningful way.

We speculated that the vmPFC may track the associative strength between Self- and Other-attributed belief updates. To test this idea directly, we have now performed a new fMRI analysis using a model-derived estimate of the probability of seeing a shared trial. We found this signal is encoded in vmPFC (and temporal pole). It seems likely that vmPFC is tracking the relationship between Self-attributed experiences and Other-attributed experiences, with downstream effects on the coding of prediction errors.

There is more information about this new analysis in our response to reviewer #2 and a new main figure (Fig. 5). We feel that this new analysis goes some way to connect the MT result with the other findings.

2. The relationships between the tasks is interesting. However, they also cast doubt on the specificity of the initial finding about plasticity of learning about self and other belief updates. The perspective taking task suggests the result is not specific to learning and the temporal discounting result suggests it is not specific to social cognition. This is particularly a concern because the identified regions, too, are not very much related to either social cognition or learning. It raises the possibility that the investigated phenomenon is much less specific than the lambda parameter (indicating self-other misattribution of PEs) suggests.

We appreciate the reviewer's concerns here.

Firstly, we'd like to clarify that we agree with the reviewer; we actually think this is one of the most interesting aspects of our results. The fact that learning about agents transfers to the perspective-taking task indicates that learning about Self-Other boundaries is *general* with respect to the task (trained on a learning task but tested in a perspective-taking task) but *specific* to the agents.

We have tried to make this point clearer by adding some text to the discussion:

Lines 433 - 438: The FBT training was general enough to affect behaviour in an independent cognitive task. Both tasks involved the same social agents, but whilst the training task required subjects to track beliefs about numerical probabilities, the transfer task required subjects to count objects in a visual scene from different perspectives. Our results suggest that the relational learning that occurred in FBT training was not specific to learning signals, but was general enough to impact on non-learning related decision variables.

We also agree that the computations we are investigating appear to *not* be specific to social cognition. We believe this is a positive feature of our results, rather than a weakness. We do not want to claim that the plasticity is specific to social cognition. Rather, we think it is likely to be a domain-general form of relational learning that is *necessary* for learning about Self-Other boundaries. In the discussion we have expanded the material about whether or not the phenomenon should be considered social per se.

Lines 494-541: It is important to consider whether our FBT engages cognitive processes that are social per se. The task does not emulate a natural social environment; there is no back-and-forth dyadic interaction, nor do subjects observe the behaviour of other agents. In essence, the task merely requires subjects to track two random variables, and perhaps the minimally social nature of the task has no bearing on the cognitive processes at play.

However, we have previously provided evidence indicating subjects are indeed sensitive to the social context of the task. We showed previously that the degree to which sensory PEs are encoded in agent-specific neural activity patterns depends on the social nature of the cover story²³. In this prior experiment, we compared the social task with a non-social task, where for the latter there was no other actual participant. Instead, subjects imagined themselves in a counterfactual situation in which they were the one

exploiting the corrupted stream of information. The social and non-social tasks were structurally identical, differing only in their cover stories and this manipulation alone was sufficient to modulate the extent to which Self- and Other-attributed PEs were neurally distinct. Furthermore, in the current study we show that training subjects on the Hi- and Lo-Share contexts of the FBT induces behavioural change in a visual perspective-taking task, suggestive that the FBT does tap into social processes.

Despite the above considerations there remains a possibility that neither the FBT nor the perspective-taking task engages social cognition. Thus, a behavioural transfer from one task to the other may simply reflect a form of non-social learning⁵³ that might be evident even if alternative versions of the tasks were conducted, for instance using arrows with different visual features as opposed to avatars with different faces. It is difficult to assess what makes a task ‘social’. Whilst we cannot say with absolute certainty that our subjects engaged in computations exclusive to socially interactive settings, we nevertheless consider these computations are likely to be co-opted when attributing mental states to social agents. For instance, being able to represent multiple models of the same environment may be a necessary component of social cognition, whilst also useful in non-social situations. Our results show that information about agent-identity and relationships between different agents’ mental states can be encoded in fundamental sensory processing signals. Whilst these signals in isolation are not ‘social’ per se, they appear to contribute to complex social cognitive processes.

The ability to learn relationships between different agents’ computations may be just one example of a form of relational learning, that is not social per se. Relational learning allows organisms to represent the world efficiently. By representing environments in terms of abstract ‘concepts’⁵⁴, ‘task sets’⁵⁵ or ‘cognitive maps’⁵⁶, animals can rapidly generalise a structure learned in one environment to a totally new environment^{57,58}. The vmPFC has also been associated with mapping latent, contextual states of the environment, in non-social situations^{59,60}. Agent identity may simply be one example of a latent environmental state, that shapes learning and behaviour to suit the current context.

Consistent with the notion that common computations can be used in both social and non-social contexts, we found that behavioural and neural measures of Self-Other distinction are related to discounting behaviour in an intertemporal choice task. Subjects who discounted future rewards more steeply also represented other agents’ mental states more distinctly from their own mental states, and were better able to distinguish the beliefs of Self and Other. This finding is consistent with a common relational learning process regulating a generalisation between Self-attributed mental states and mental states attributed to both other agents and to one’s future Self. It is also consistent with prior theoretical accounts that propose a common mechanism for traversing social and temporal distances^{33-35, 61}.

3. Behavioural modelling: Along the lines of the previous point – the lambda parameter is perhaps the most important parameter in the model because it relates to the degree that PEs are mis-assigned to the wrong agent which sets the rationale for the PE merging MRI decoding. However, each of the 4 sessions has a different best fitting model casting doubt on whether the lambda indeed the critical variable.

Can the authors show that: • The differences in correlated beliefs (Fig 2b) do not emerge from models without the lambda (using model simulations and the next-best fitting models that don't contain a lambda)?

We thank the reviewer for this helpful suggestion. We have repeated the analysis from Fig 2b but using the next-best fitting models that don't contain lambda parameters. We find that the effect is absent if these models are used (Supplementary Fig. 3a)

We then ran model simulations, modulating the value of lambda in each subject to see how the effects shown in Fig 2b are sensitive to both increases and decreases in lambda (Supplementary Fig. 3b). We find that increasing lambda above subjects' true values boosts the effect, and reducing lambda reduces the effect.

The new supplementary figure is shown below.

Supplementary Fig. 3 | Simulations show that λ is a determinant of the observed behavioural effects. *a*, The analysis shown in Fig. 2b was repeated but Self- and Other-attributed beliefs were estimated using the next best-fitting models, that contained no λ parameters. These were models 4 (Lo-Share) and 64 (Hi-Share) for the training session, and model 13 (both conditions) for the testing session. When beliefs were estimated using these models, there was no difference, in training or testing, between the Hi- and Lo-Share conditions, in Self-Other correlation. *b*, The analysis shown in Fig. 2b was repeated, using the winning models, but each subject's λ parameter in the Hi-Share context was modulated. When the analysis was conducted using increased λ values (red lines), the effect shown in Fig. 2b became stronger. When the analysis was conducted using reduced λ values (blue lines), the effect shown in Fig. 2b became weaker. The range of modulations tested was -0.006 to +0.006.

- Analogous to the parameter recovery in Fig. 2d, can model recovery based on model simulations using a confusion matrix reliably identify their winning models? Given that overall 72 models were fitted to the data, this seems to be the more pertinent control compared to the parameter recovery analysis.

We have added a new supplementary figure with a model recovery analysis. For computational feasibility and interpretability for readers, we only included the four best-fitting models from the four datasets in the model recovery analysis. Although there is some trade-off between models 4 and 13, and between 65 and 68, none of our hypotheses or conclusions rest on being able to distinguish between these models. Rather, we need to be able to distinguish between models that do contain the leak parameter and models that don't contain the leak parameter. We find that we can do this well. The methods and figure are shown below:

Lines 678-687 (Methods): Model recovery was performed by simulating choice data using each of the four best-fitting models for the four datasets, using the parameters estimated for each subject. Each of these four models was then fit to each of these four simulated datasets. For each simulated dataset, we computed the proportion of subjects for whom each of the four models was the best-fitting (lowest BIC). This gives us $p(\text{fit}|\text{sim})$, the probability that a model is best-fitting, given that another (or the same model) simulated the data. $P(\text{fit}|\text{sim})$ was converted into $P(\text{sim}|\text{fit})$ using Bayes' rule as follows.

$$P(\text{sim}|\text{fit}) = \frac{p(\text{fit}|\text{sim})p(\text{sim})}{\sum_{\text{sim}} p(\text{fit}|\text{sim})p(\text{sim})}$$

We assumed a uniform prior on models. This technique to derive $P(\text{sim}|\text{fit})$ is described in more detail in a recent review by Wilson and Collins⁶³.

Supplementary Fig. 4 | Model recovery. **a**, We simulated FBT choice data using each of the 4 best fitting models from for the 4 datasets, using the parameters estimated for each subject. We then fit all 4 models to these simulated data. First, we computed $P(\text{fit}|\text{simulated})$ as the probability that a model was the best-fitting of the 4 models, given that it was the generative model. Then, we used Bayes' rule to compute $P(\text{simulated}|\text{fit})$ as the probability that a model was the generative model, given that it was the best-fitting of the 4 models. This 'inversion matrix' is shown. For example, the 0.79 in the third row and third column, indicates that if model 13 was the best-fitting, of the four models, then there is a 0.79 chance that this model truly generated the data. See Methods for further details. **b**, We noted that there was a trade-off between models 4 and 13, and also between models 65 and 68. Given that we were primarily interested in being able to distinguish models that contain λ and models that do not contain λ , we repeated the model recovery analysis, but computed the probability that one of the λ -containing models generated the data, given that either of them was the best-fitting model (0.96) and the probability that one of the non- λ -containing models generated the data, given that either of them was the best-fitting model (0.9). This recovery was more successful.

- The lambda effects persist when controlling for overall performance and decision noise?

We have now adjusted the leak factor ($\lambda : \alpha$) throughout the paper by regressing out both decision temperature (τ parameter) and overall performance. We have corrected the statistics reported for the correlation between leak factor and $\Delta(\text{cross-decoding})$, as well as correlation between leak factor and discount factor, in the main text, accordingly. These effects remain significant.

Minor points: 1. I am sceptical whether the visual perspective taking result really maps onto the FBT experiment. The authors make the point in the discussion that “ we observed agent-specific transfer effects on both congruent and incongruent trials.” However, supplementary Fig. 6 looks as if RT and accuracy benefits are mostly in the congruent condition, which does not map very well on the FBT results. I understand that the conclusions are derived from the drift diffusion model that modelled several drift rates for different conditions relative to the control condition. However, I could not work out a) which ones of those are shown in Fig.3c and b) whether panel 3c indeed shows evidence for separate drift rate effects relating to both congruent and incongruent trials.

Reaction time and accuracy effects were present in both congruent and incongruent trials. These effects were indeed stronger on congruent trials than incongruent trials, but this was not a significant difference – i.e. there was no significant interaction between congruency and avatar on corrected drift rate change [repeated measures 2-way ANOVA: $F(1, 45) = 0.28, P = 0.6$]. We apologize this wasn't clear from the original figures. We have now updated Figure 3c (reproduced below). Where we previously showed averages across congruent and incongruent trials, we now show congruent and incongruent separately. Figure 3b still averages over Lo-Share and Hi-Share drift rates because this panel is meant to illustrate the incongruency effect in the perspective-taking task.

2. Task performance is measured by the correlation of participants predictions with the true Bernoulli parameter. While there is some plausibility to this measure as an index of performance, it ignores some aspects of the predictions like the range or the intercept and it also deviates from the authors own measure of performance in previous paper on this paradigm. Why was this measure chosen and do the results in fig.2a hold for other measures of performance such as the one used previously or the one implicit in the RL model?

We opted for this measure of performance because we thought it was more intuitive than the measure we used in our previous paper (deviation from ground truth). However, the results in fig 2a are virtually identical between the two measures (and are statistically significant with both measures). The figure below reproduces Fig. 2a, but using our old accuracy metric. We have not included this in the paper but we are happy to do so if requested.

3. The fMRI regressors are derived from the more complex RL model which fits only one of the two MRI sessions best. Why was this model chosen – rather than for example the model that fits best when averaging over both of the sessions? That seems to allow for a fairer comparison between the session types. Could the differences in MRI self-other decoding be related to this modelling choice?

We wanted to only use a single set of parameters for the entire fMRI analysis, to avoid introducing systematic differences between the Lo-Share and Hi-Share conditions. We chose the parameters from the complex model because the simple model is nested within the complex model.

We in fact derived the parameters for the fMRI analysis by averaging across the Lo- and Hi-Share sessions. We acknowledge that this was unclear in our Methods section, and thank the reviewer for identifying this. This means that we accommodated the behavioural findings of both sessions (e.g. low evidence of self-other leakage in Lo-Share, but high evidence of self-other leakage in Hi-Share), whilst also maximising our ability to explain signal across the fMRI sessions. We thought this was a reasonable compromise. We have added text to our methods to make sure that this is clearer:

Lines 669-673: For the fMRI analysis in the test session (day 3), the same model and the same parameters were used for each subject to estimate PE regressors. The more complex of the two winning models (for Lo-Share and Hi-Share) was selected (M68). For each subject, we averaged the parameters across the two sessions, and then took the median parameter values across subjects.

We believe that the differences in MRI self-other decoding cannot arise from this modeling choice. If our choice of parameters were biased to better explain data in one session than the other, we would end up with better decoding in Hi-Share in both the pseudo-trial *and* the cross-decoding analysis. A bias in favour of one session should not be able to produce the double dissociation we observe, where Self/Other decoding is higher in the Lo-Share context, but cross-decoding is higher in the Hi-Share context.

4. Typo page 3 line 54: “recircuit”

In light of the comments from reviewer #1 we have replaced all instances of the word recircuit with the word “reconfigure” instead.

5. Personally, I don't find the summary figure very useful or intuitive. This comment is not about the science behind the paper and I would leave it up to the authors if they want to change it or not. I would guess that changing that figure can help convey the author's findings to a broader audience.

We have removed this figure from the paper.

Reviewers' comments:

Reviewer #1 (Remarks to the Author):

The authors have generally addressed the issues I raised in my initial review. I appreciate the clarification that there was in fact a real social partner involved - that was not clear to me from the my initial reading. That fact does substantially improve the face validity of the paradigm. I still think we should be cautious in generalizing these results to more naturalistic contexts, but the authors have sufficient addressed that in the text. It is also reassuring to see that TPJ and MPFC activation arise from some reasonable contrasts. The use of GFRT is not ideal in my mind - Eklund et al. still showed substantially elevated false positives using this correction in at least one of their two datasets when using a $p < .001$ voxelwise threshold. However, I acknowledge that the results are likely to be qualitatively similar either way. I think this article could now make a valuable contribution to the literature without further major changes.

Reviewer #2 (Remarks to the Author):

Ereira et al revised the manuscript abundantly. I particularly appreciate the efforts the efforts made, but I am still disappointed by the lack of a unifying modeling framework.

I have a remaining important concern about the model-based fMRI analyses and results. A recent paper (Lebreton et al., 2019, Nature Human Behaviour) clearly illustrated that the definition and preprocessing of the model-based regressors significantly affect inter-individual differences results. Even worst, they can create spurious inter-individual correlations and results. The authors should refer to this paper and its guidelines and provide more information about the definition and preprocessing of the variables. They should also check if the scaling issues mentioned in the papers can affect their results.

Another concern is that the larger the cohorts (e.g., human brain project), the smaller the effect size reported in structural imaging. Authors should also report what was the statistical power of their structural MT analyses/results. How their sample size compare to other MT studies?

Reviewer #3 (Remarks to the Author):

The authors have revised their manuscript and added some new analyses that partly address my concerns. From my perspective, the RL modelling work has been strengthened quite a bit by the simulation work showing that 1) belief overlap between self and other is related to the lambda parameter rather than other features of the model and that 2) models with a lambda parameter can be reliably dissociated from one without lambda. I like the new univariate MRI analysis that looks at updates of the probability of shared trials which indeed seems tightly related to the behavioural pre-

training effect that is very interesting. The analysis identifies brain regions (temporal pole and vmPFC) that make sense based on the what we know about these brain networks. The weakness that the main fMRI analysis lacks any anatomical specificity remains and there was no other attempt by the authors to address this point apart from the above analysis that is at least indirectly related. As such, I still think the results don't add very much to the authors previous paper in terms of neural findings regarding self-other distinction. I take the point that one study is an intervention study and the other one is not. That does not change much regarding the neural conclusions though. However, overall, I feel that the authors have produced a thorough and interesting behavioural and computational investigation of social learning that particularly given the huge effort that went into setting up this sort of work would be worthwhile publishing in this journal as such.

Blue = reviewer comments

Black = response to reviewer

Ereira et al revised the manuscript abundantly. I particularly appreciate the efforts made, but I am still disappointed by the lack of a unifying modelling framework.

As is standard in the field, for each task domain we have used an established and well-understood modelling framework. Although we agree that unifying the tasks within a single modelling framework would be an interesting exercise, we remain of a view that this would address a different research question from the one we seek to address here. In fact, this enterprise would mandate an entire research article in its own right.

I have a remaining important concern about the model-based fMRI analyses and results. A recent paper (Lebreton et al., 2019, Nature Human Behaviour) clearly illustrated that the definition and preprocessing of the model-based regressors significantly affect inter-individual differences results. Even worse, they can create spurious inter-individual correlations and results. The authors should refer to this paper and its guidelines and provide more information about the definition and preprocessing of the variables. They should also check if the scaling issues mentioned in the papers can affect their results.

Thanks for letting us know about this insightful paper from Lebreton and colleagues. The Lebreton et al. paper details how measures of BOLD activation (regression weights) are inversely proportional to the individual variance of the explanatory behavioural variable, and consequently inter-individual brain behaviour differences (IBBD) are sensitive to whether regressors in a GLM are z-scored or native (not z-scored). Specifically, under proportional-coding assumptions, activations from native regressors will show no inter-individual differences, whereas activations from z-scored regressors will show inter-individual differences. However, under range-coding assumptions, inter-individual differences in activation measures from native regressors will inversely scale with behavioural differences, whereas activations from z-scored regressors will not vary across individuals. This raises interpretability issues for IBBB results.

In our paper we conduct one IBBB analysis, which is shown in Figure 4d. Here we correlated an fMRI measure (difference in cross-decodability between the two sessions) with a leak factor, a trait behavioural variable. The fMRI measure here is not an 'activation measure'. Rather, it derives from decoding accuracies, i.e. to what extent can a linear regression model accurately predict PE magnitude from multi-voxel BOLD data? This variable satisfies Lebreton et al.'s recommendation of using a measure of "how well" a variable is encoded in the BOLD signal rather than "how much". Whilst we used a GLM as a feature selection step, to identify a group of voxels for the decoding analysis, we never used participants' activations as a variable for exploring interindividual variability.

However, in Figure 5c, we *did* report an analysis where we explored intersubject variability in BOLD activations, specifically regression weights from z-scored

regressors. In this analysis we tested for a between-subjects correlation between BOLD activation associated with the perceived probability of encountering a shared trial and the difference in cross-decoding between the two sessions. This is not an IBB analysis; we were not looking for an association between BOLD activation and a behavioural trait, but rather a relationship between BOLD activation and the cross-decoding measure from a previous fMRI analysis. A priori, we were not sure whether this kind of analysis would be sensitive to the kinds of scaling issues that Lebreton et al. describe. Therefore, as requested by the reviewer, we re-ran this analysis using native (not z-scored) regressors.

Reassuringly, we replicated all the findings from Figure 5. With the 0.01 learning rate, the vmPFC cluster remained significant (194 voxels, $P_{FWE} = 0.006$, small-volume corrected for cluster extent, peak co-ordinates: $x = 3, y = 46, z = -15$). The intersubject correlation shown in figure 5c was still present ($r = 0.33, P = 0.04$). With a 0.025 learning rate, the temporal pole cluster remained significant (630 voxels, $P_{FWE} < 0.001$, cluster level, peak co-ordinates: $x = -62, y = -16, z = -16$). The correlation between temporal activation and delta(cross-decoding) was again, non-significant ($r = -0.04, p = 0.82$).

We have added the following text to the Methods section:

Lines 790-791: All regressors were z-scored within subjects.

Lines 919-924: In this analysis, the model-based regressor was z-scored within each subject, before estimating GLMs and each subject's BOLD activation. We note that it has been shown recently that inter-individual variability in BOLD activation is highly sensitive to the way that regressors pre-processed, particularly whether they are z-scored or not⁷⁰. We re-ran this analysis using the native P regressor, without z-scoring it, to assess whether our results were sensitive to this pre-processing step. Reassuringly, this re-analysis replicated all results reported in Fig. 5.

Another concern is that the larger the cohorts (e.g., human brain project), the smaller the effect size reported in structural imaging. Authors should also report what was the statistical power of their structural MT analyses/results. How does their sample size compare to other MT studies?

'Big data' projects such as the Human Brain Project usually measure macrostructure parameters, such as grey and white matter volume. By using MT, a measure of brain microstructure, we expect to have higher sensitivity for detecting interindividual variability. It's also important to note that big data projects naturally exhibit lower effect sizes due to the additional variance introduced by factors such as different scanners, scanning sites and experimenters. These biases are absent from smaller projects such as ours.

Previous studies investigating intersubject variability in MT maps have used sample sizes ranging from 31-50 (Steiger et al. 2016, Allen et al. 2017, Palaniyappan et al. 2018, Geeraert et al. 2019). To assess the power of our MT analysis, we looked at two of these studies that ran the same statistical test that we did, namely a whole-

brain second-level regression on MT maps with a single covariate of interest (Allen et al. 2017 and Steiger et al. 2016). Using Pearson correlation coefficient as effect size, these studies reported effect sizes of $r = 0.62$ (48 participants) and $r = 0.69$ (31 participants), respectively. We determined that to detect an effect size of $r = 0.64$ with a type I error rate of $\alpha = 0.001$ (cluster-forming threshold) and a type II error rate of $\beta = 0.1$ (90% power), we would need a sample size of 39 participants. As our sample size was 40, we can have confidence that the analysis is adequately powered (in excess of 90%) to detect effects of similar magnitude to those seen in these previous studies.

We have added the following text:

Lines 892-902: To ensure our analysis had adequate statistical power to detect interindividual variability in MT maps, we looked at effect sizes from previous studies that conducted regression analyses on MT maps. Allen et al. reported a peak voxel effect of $r = 0.62$ with a sample size of 48 participants, when correlating metacognitive ability with MT in a whole-brain analysis⁶⁹. Steiger et al. reported a peak voxel effect of $r = 0.69$ with a sample size of 31 participants, when correlating memory ability with MT in a whole-brain analysis⁷⁰. To detect an effect size (correlation) of 0.64 with a type I error rate of $\alpha = 0.001$ (cluster-forming threshold) and a type II error rate of $\beta = 0.1$ (90% power), we would need a sample size of 39 participants⁷¹. As our sample size in this analysis was 40, we consider we are adequately powered to detect effects of similar magnitude to those seen in previous studies of interindividual MT variability.

Reviewer #2 (Remarks to the Author):

I have no additional remark.